# Influence of Voluntary Contraction Level, Test Stimulus Intensity and Normalization Procedures on the Evaluation of Short-Interval Intracortical Inhibition

**DOI:** 10.3390/brainsci10070433

**Published:** 2020-07-08

**Authors:** Cécilia Neige, Sidney Grosprêtre, Alain Martin, Florent Lebon

**Affiliations:** 1INSERM UMR1093-CAPS, Université Bourgogne Franche-Comté, UFR des Sciences du Sport, F-21078 Dijon, France; alain.martin@u-bourgogne.fr (A.M.); florent.lebon@u-bourgogne.fr (F.L.); 2EA4660-C3S Laboratory—Culture, Sport, Health and Society, Université Bourgogne Franche-Comté, 25000 Besançon, France; sidney.grospretre@univ-fcomte.fr

**Keywords:** SICI, muscle contraction, transcranial magnetic stimulation, corticospinal excitability

## Abstract

Short-interval intracortical inhibition (SICI) represents an inhibitory phenomenon acting at the cortical level. However, SICI estimation is based on the amplitude of a motor-evoked potential (MEP), which depends on the discharge of spinal motoneurones and the generation of compound muscle action potential (M-wave). In this study, we underpin the importance of taking into account the proportion of spinal motoneurones that are activated or not when investigating the SICI of the right flexor carpi radialis (normalization with maximal M-wave (Mmax) and MEP_test_, respectively), in 15 healthy subjects. We probed SICI changes according to various MEP_test_ amplitudes that were modulated actively (four levels of muscle contraction: rest, 10%, 20% and 30% of maximal voluntary contraction (MVC)) and passively (two intensities of test transcranial magnetic stimulation (TMS): 120 and 130% of motor thresholds). When normalized to MEP_test_, SICI remained unchanged by stimulation intensity and only decreased at 30% of MVC when compared with rest. However, when normalized to Mmax, we provided the first evidence of a strong individual relationship between SICI and MEP_test_, which was ultimately independent from experimental conditions (muscle states and TMS intensities). Under similar experimental conditions, it is thus possible to predict SICI individually from a specific level of corticospinal excitability in healthy subjects.

## 1. Introduction

Cortical inhibitory and excitatory circuits within the primary motor cortex (M1) play an important role in the fine tuning of descending motor commands and in the neural adaptation following motor training [1]. For example, the reduction in intracortical inhibition within M1 is a crucial part of neural adaptation following acute and multi-session challenging strength training programs [2,3,4]. Paired-pulse transcranial magnetic stimulation (TMS) enables the non-invasive examination of these intracortical excitatory and inhibitory circuits at the time of the stimulation [5,6]. Short-interval intracortical inhibition (SICI) is a well-established paired-pulse measure to evaluate inhibitory circuits within the M1 area. SICI results from a subthreshold conditioning stimulus (CS) followed 1 to 6 ms later by a suprathreshold test stimulus (TS) applied through the same coil over M1 [5]. The high-intensity suprathreshold TS leads to a corticospinal output called motor-evoked potential (MEP), measured peripherally by means of electromyographic (EMG) recording over the target muscle. With optimal settings, the CS exerts a decrease in the conditioned MEP (MEP_cond_) amplitude in comparison with the unconditioned MEP (MEP_test_) induced by a TS alone [5]. It is thought that the CS activates low-threshold inhibitory interneurons that employ γ-aminobutyric acid type A receptor (GABA_A_) and reduce the excitatory inputs activated by the TS [5,7,8]. The amount of SICI is conventionally quantified by expressing the mean MEP_cond_ amplitude as a percentage of the mean MEP_test_ amplitude [5] or the difference between the mean MEP_cond_ and the mean MEP_test_ then expressed as a percentage of the mean MEP_test_ amplitude (SICI_MEPtest_) [9]. An issue with this normalization is that the size of the MEP_test_ is known to influence SICI outcome. Indeed, previous studies have shown that in resting hand muscles, the amount of SICI_MEPtest_ increased for MEP_test_ amplitude up to 1 mV, but further remained unchanged for larger MEP_test_ amplitude, with a trend to a linear relationship between the MEP_cond_ and MEP_test_ [9,10,11,12].

Importantly, it is widely accepted that SICI represents an inhibitory phenomenon acting purely at the cortical level (rather than subcortical or spinal levels); indeed, the CS does not alter the spinal excitability when delivered at low stimulation intensities (<80% of the motor threshold) and measured in a static muscle [5,13], as evidenced by epidural recordings [7]. However, one has to keep in mind that SICI estimation is based on the MEP_cond_ amplitude recorded peripherally, and therefore also depends critically on the discharge of spinal motoneurones and the generation of a compound muscle action potential [9,14]. Taking into account this important point, Lackmy and Marchand-Pauvert (2010) assessed SICI_Mmax_ by calculating the difference between the mean MEP_cond_ and the mean MEP_test_, then expressed as a percentage of maximal compound muscle action potentials (Mmax (M-wave)). Mmax is evoked when all the motor axons of the target muscle are activated simultaneously by a supramaximal peripheral nerve stimulation. SICI_Mmax_ normalisation has the methodological benefit of considering the proportion of spinal motoneurones activated by the TS. Interestingly, Lackmy and Marchand-Pauvert (2010) found increased SICI_Mmax_ with increasing TS intensities (i.e., increased MEP_test_/Mmax ratio) [9]. However, this work has only focused on resting hand muscle, whereas muscle activation is known to decrease the amount of SICI_MEPtest_ [15,16]. To fill this gap, Opie and Semmler (2014) subsequently tested whether SICI_Mmax_ (calculated with MEP_test_ expressed relative to Mmax) was influenced when the muscle was voluntary activated. Contrary to SICI_MEPtest,_ they found that the amount of SICI_Mmax_ during the active state increased concomitantly with an increase in MEP_test_/Mmax ratio [14]. Importantly, larger MEP_test_/Mmax ratios were obtained by increasing the TS intensity (from 110% until 150% of the motor threshold) during the active state. However, in this study, active muscle condition was assessed with only one low level of contraction (5% of maximal voluntary contraction (MVIC)), whereas SICI_MEPtest_ has been shown to vary with increasing level of muscle contraction and MEP_test_ amplitude [10,16,17,18,19]. It is not yet known whether SICI_Mmax_ is modulated when assessed at higher levels of muscle contraction.

The aim of the present study was to broaden current knowledge on the necessity of considering the proportion of spinal motoneurones activated by single- and paired-pulse TMS in different muscle states. We quantified SICI with varying MEP_test_ amplitudes modulated actively (with four levels of muscle contraction) and passively (with two TS intensities of stimulation). The hypothesis was that SICI_Mmax_ would increase in response to larger MEP_test_/Mmax ratios obtained by an increasing muscle contraction force. By considering this prediction as well as Opie and Semmler’s (2014) findings, we also hypothesized that the amount of SICI_Mmax_ would be highly dependent upon the proportion of motoneurones activated by single-pulse TMS, regardless of the conditions leading to MEP_test_ increase.

## 2. Materials and Methods

### 2.1. Participants

Fifteen healthy volunteers (six females; mean age 24.9 years, range 22–30 years; right-handed as assessed by the Edinburgh Handedness Inventory [20]) participated in the current study after providing written informed consent. All volunteers were screened by a medical doctor for contraindications to TMS [21] and were excluded if they had a history of psychiatric or neurological illness. The protocol was approved by the “Comité de Protection des Personnes Sud-Ouest et Outre Mer III” (CPP SOOM III) ethics committee (number 2017-A00064-49; ClinicalTrials.gov Identifier: NCT03334526) and complied with the Declaration of Helsinki.

### 2.2. General Experimental Procedure

Participants came to the laboratory for a single 1.5-h session and were seated in an isokinetic dynamometer chair (Biodex System 3, Biodex Medical Systems Inc., Shirley, NY, USA). The participants’ right hand was firmly strapped in a neutral position to a custom-built accessory adapted for wrist movement recordings. The rotation axis of the dynamometer was aligned with the styloid process of the ulna. The upper arm was vertical along the trunk (shoulder abduction and elevation angles at 0°) and the forearm semipronated and flexed at 90° (Figure 1). Before starting the main experiment, participants familiarized themselves with the procedure of the voluntary force production feedback during an approximately 5 min warm-up of wrist flexions. They received visual feedback of all exerted force contraction on a computer screen located 1-m in front of them. Participants then performed 3 maximal voluntary isometric contractions (MVIC) lasting 3 s with at least 30 s rest in between. Verbal encouragement was provided during MVIC trials to achieve a maximal force. The maximum of the three trials was defined as the participant’s MVIC. This value was then used to calculate the muscle contraction levels (i.e., 10, 20 and 30% MVIC) tested in the current study.

For each level of contraction (rest, 10%, 20% and 30% MVIC), three Mmax traces, twelve single-pulse MEPs (MEP_test_) and twelve paired-pulse MEPs (MEP_cond_) delivered with each of the TS intensities (120% and 130% of motor thresholds (MT)) were recorded per subject. The order of the different levels of contraction was randomized across participants. Single-pulse and paired-pulse TMS measures obtained with the two TS intensities performed in different recording blocks and the order of the blocks randomized across participants. The inter-trial intervals between single-pulses were at least 3 s [22]. During active conditions, the mean duration of the recording was set at 500 ms and participants were asked to relax after the stimulation for few seconds before the next trial and contracting again. This was done to avoid fatigue generated by holding the level of contraction across all consecutive trials.

### 2.3. EMG Recordings

After shaving and dry-cleaning the skin with alcohol, surface electromyographic (EMG) recordings of the right flexor carpi radialis (FCR) were made with two silver-chloride (Ag/AgCl) electrodes placed over the muscle belly with an interelectrode center-to-center distance of 2 cm, at 1/3 of the distance from the medial epicondyle of the humerus to the radial styloid process (Figure 1). A ground electrode was placed over the medial epicondyle to the radial styloid. The EMG signals were amplified (gain: 1000) and band-pass filtered (10–1000 Hz), digitized at a sampling rate of 2000 Hz and stored for off-line analysis (Biopac Systems Inc., Goleta, CA, USA).

### 2.4. Electrical Stimulation Protocol

To evoke Mmax, 1-ms square wave pulses were delivered over the median nerve near the cubital fossa via bipolar felt pad electrodes (8-mm diameter with 30-mm spacing) (Figure 1). The stimulation site, defined as the location that produced the largest Mmax, was first determined, then the electrodes, connected to a constant-current stimulator (DS7R, Digitimer, Welwyn Garden City, Hertfordshire, UK), were fixed at the arm by a Velcro strap. The intensity of stimulation was gradually increased until the amplitude of the M-wave did not increase any further. Then, 120% of this last intensity was used for Mmax recording, in order to ensure that the response lay in the plateau of its maximal value (mean intensity: 17.71 mA; range 13.2–28.2 mA).

### 2.5. Transcranial Magnetic Stimulation Protocol

Single-pulse and paired-pulse stimulations were applied with a 70-mm figure-of-eight coil connected to a monophasic Magstim BiStim^2^ stimulator (The Magstim Co., Whitland, UK). The coil was placed over left M1, tangentially to the scalp with the handle pointing backward and laterally at 45° away from the midsagittal line, resulting in a postero-anterior current flow within M1. The optimal stimulation site on the scalp (hotspot) was defined as the location eliciting the largest MEP amplitude in the FCR muscle for a given intensity. This location was marked by a color marker on a tight-fitting cap worn by the participant. Because the aim of the current study was to compare the effect of different levels of muscle contraction (i.e., rest, 10, 20 and 30% of MVIC) and two TS intensities (i.e., 120% and 130% of the motor thresholds (MT)), MT were systematically calculated according to these different conditions. The resting motor threshold (rMT) was determined as the lowest stimulus intensity required to evoke at least 5 MEPs of 50 μV peak-to-peak amplitude out of 10 consecutive trials in the relaxed muscle [23]. For the 10%, 20% and 30% MVIC, the active motor threshold (aMT) was determined as the lowest stimulus intensity required to evoke at least 5 MEPs of 200 μV peak-to-peak amplitude out of 10 consecutive trials [23]. Some participants (*n* = 4) presented an EMG level greater than 200 μV during muscle contraction. The aMT was then determined as the lowest stimulus intensity required to evoke at least 5 MEPs out of 10 consecutive trials with a peak-to-peak amplitude greater than the 95% confidence interval of the prestimulus mean EMG activity for each contraction level [16]. For SICI measurements, the CS intensity was set at 70% rMT for rest conditions and 70% aMT for active conditions [9,16]. The inter-stimulus interval between TS and CS was set at 3 ms [5].

### 2.6. Data Analysis

Data analysis was performed using the Acqknowledge software package (Version 4.2, Biopac Systems Inc., Goleta, CA, USA).

The background root mean square of the surface EMG (EMG_rms_) of the right FCR was calculated during the 100 ms epoch prior to the TMS artefact in order to quantify the pre-stimulus neuromuscular activation for all conditions. EMG_rms_ activity was normalized using the corresponding Mmax amplitude by the following ratio (EMG_rms_/Mmax). EMG_rms_ activity was also normalized by using the maximal EMG_rms_ obtained during the participant’s MVIC.

Responses were recorded in the right FCR muscle. The peak-to-peak amplitudes were measured for Mmax, MEP_test_ and MEP_cond_ in each experimental condition. Figure 2 shows typical raw Mmax (2A), MEP_test_ (2B, white panel) and MEP_cond_ (2B, grey panel) elicited at 120% MT and MEP_test_ (2C, white panel) and MEP_cond_ (2C, grey panel) elicited at 130% MT.

In order to investigate how the estimation of SICI is influenced according to the normalization, SICI was expressed across all conditions using the following two equations:

(1) The ‘usual’ equation provides an estimation of SICI (SICI_MEPtest_) by calculating the difference between the mean MEP_cond_ and the mean MEP_test_, then expressed as a percentage of the mean MEP_test_ [9].
(1)$$\mathrm{SICI}\;\left(\%{\mathrm{MEP}}_{\mathrm{test}}\right)=\frac{{\mathrm{MEP}}_{\mathrm{cond}}\;-\;{\mathrm{MEP}}_{\mathrm{test}}\;}{{\mathrm{MEP}}_{\mathrm{test}}}\times 100$$

(2) The second equation provides an estimation of SICI (SICI_Mmax_) by calculating the difference between the mean MEP_cond_ and the mean MEP_test_ then expressed as a percentage of the mean Mmax [9,14].
(2)$$\mathrm{SICI}\;\left(\%\mathrm{Mmax}\right)=\frac{{\mathrm{MEP}}_{\mathrm{cond}}\;-\;{\mathrm{MEP}}_{\mathrm{test}}\;}{\mathrm{Mmax}}\times 100$$

For both equations, negative values indicate inhibition and positive values indicate facilitation.

### 2.7. Statistical Analysis

All statistical analyses were performed using IBM’s SPSS software version 24 (SPSS Inc., Chicago, IL, USA). Normality of the data distributions was verified using the Shapiro–Wilk test and the assumption of normality was not violated for any of the data. Homogeneity of variances was assessed by Mauchly’s test and a Greenhouse–Geiser correction was applied if the sphericity assumption was violated. Pre-planned post-hoc analyses were performed on significant interactions after applying a Bonferroni correction for multiple comparisons. Uncorrected degrees of freedom and corrected *p* values for multiple comparisons are reported in the results section. The α level for all analyses was fixed at 0.05. Partial eta squared (η_p_^2^) values are reported when results are statistically significant in order to express the portion of the total variance attributable to the tested factor or interaction. Values are reported as mean ± SD.

A first set of analyses were performed to control for potential methodological biases. The stimulation intensities (%MSO) for the MT as well as the EMG_rms_/Mmax ratios and the Mmax amplitude obtained for each level of contraction were compared using a one-factor analyses of variance (ANOVA).

The normalized TMS data (MEP_test_/Mmax; SICI_MEPtest_ and SICI_Mmax_) were separately analysed with repeated-measure ANOVA with two within-subject factors: TS intensity_2_ (120% rMT or aMT vs. 130% rMT or aMT) and levels of contraction_4_ (i.e., rest, 10, 20 and 30% of MVIC).

In order to explore how SICI_Mmax_ can be predicted from the MEP_test_/Mmax ratio, a regression coefficient analysis was estimated individually for each participant, according to procedures described by Pfister and colleagues [24]. Individual slopes, intercept, R^2^ and *p* values were extracted and reported.

Then, to investigate how SICI_MEPtest_ and SICI_Mmax_ are influenced by the MEP_test_/Mmax ratio that is, regardless of the level of contraction or the TS intensity, results were reorganised by pooling the MEP_test_/Mmax ratio into 5% bins, which resulted in bins from 0–5% to 45–50% and then into 10% bins (50 to 60% and 60 to 70%) or a percentage corresponding to >70%. This also allowed to take into account inter-individual variability. Mean regression coefficients were calculated and reported for the two SICI normalizations.

## 3. Results

### 3.1. Methodological Considerations

The one-factor ANOVA conducted to compare the MT (rMT vs. aMT during 10% MVIC vs. 20% MVIC vs. 30% MVIC) showed a significant main effect (F_(3,42)_ = 24.59; *p* < 0.001; η_p_^2^ = 0.637, Table 1). Post-hoc comparisons revealed that only the rMT differs from all aMT (all *p* < 0.002). Post-hoc comparisons also indicate that, although systematically calculated for each level of contraction, the aMT did not significantly differ depending on the level of muscle contraction. An ANOVA performed on the mean Mmax amplitude obtained for each level of contraction showed no significant main effect (F_(3,42)_ = 1.153; *p* = 0.323, Table 1), indicating that the increase in the level of muscle contraction from rest to 30% MVIC did not affect Mmax amplitude.

### 3.2. Neurophysiological Data

Figure 3 illustrates the EMG_rms_/Mmax ratios pooled for single and paired pulses and the two TS intensities (120% and 130% MT). The ANOVA carried out on these EMG_rms_/Mmax ratios revealed a significant main effect of the levels of contraction (F_(3,42)_ = 83.32; *p* < 0.001; η_p_^2^ = 0.856). Post hoc analyses showed a lower EMG_rms_/Mmax ratio for the rest condition compared with the 10% MVIC (*p* < 0.001), 10% MVIC compared with 20% MVIC (*p* = 0.003) and 20% MVIC compared with the 30% MVIC (*p* = 0.001).

Figure 4A illustrates the MEP_test_/Mmax ratios according to the TS intensity and the levels of contraction. The ANOVA revealed a main effect of the TS intensity (F_(1,14)_ = 26.97; *p* < 0.001; η_p_^2^ = 0.658) showing that the MEP_test_/Mmax ratio was greater with a TS intensity set at 130% MT than with a TS intensity set at 120% MT. A significant large effect of the levels of muscle contraction was also observed (F_(3,42)_ = 120.01; *p* < 0.001; η_p_^2^ = 0.896). Post hoc analyses showed a significant increase in the MEP_test_/Mmax for 10% MVIC compared with rest (*p* < 0.001), for 20% MVIC compared with 10%MVIC (*p* < 0.001) and for 30% MVIC compared with 20% MVIC (*p* < 0.001). The TS intensity by levels of contraction interaction was not significant (F_(3,42)_ = 2.16; *p* = 0.107).

Figure 4B illustrates the SICI when normalized to MEP_test_. The SICI_MEPtest_ did not differ according to the TS intensity (F_(1,14)_ < 1; *p* = 0.538) but the levels of muscle contraction main effect was significant (F_(3,42)_ = 5.15; *p* = 0.020; η_p_^2^ = 0.268). Post-hoc analyses showed that SICI_MEPtest_ was higher when measured at rest than during 30% MVIC (*p* = 0.043). The TS intensity by levels of muscle contraction interaction was not significant (F_(3,42)_ = 1.99; *p* = 0.161).

Figure 4C illustrates the SICI when normalized to Mmax. Similar to the ratio MEP_test_/Mmax, the SICI_Mmax_ was significantly higher with a TS intensity of 130% MT when compared with a TS intensity of 120% MT (F_(1,14)_ = 6.73; *p* = 0.021; η_p_^2^ = 0.325). A significant large main effect of the levels of contraction was found (F_(3,42)_ = 63.94; *p* < 0.001; η_p_^2^ = 0.830). Post-hoc analyses revealed that SICI_Mmax_ was higher at 10% MVIC when compared with rest (*p* < 0.001) and higher at 20% MVIC when compared with 10% MVIC (*p* = 0.006). The TS intensity by levels of muscle contraction interaction failed to reach significance (F_(3,42)_ = 1.73; *p* = 0.199).

As reported in Table 2, the regression analysis was also significant for all subjects and provided evidence for a strong linear relationship between the SICI_Mmax_ estimation and the MEP_test_/Mmax ratios. This indicates that it is possible to predict SICI_Mmax_ in an individual from a specific MEP_test_/Mmax ratio.

SICI_MEPtest_ (Figure 5A) and SICI_Mmax_ (Figure 5B) plotted according to the MEP_test_/Mmax values pooled into different bins. First, a simple linear regression was calculated on mean bins values in order to predict the SICI_MEPtest_ based on the MEP_test_/Mmax ratio (Figure 5C). A non-significant regression equation was found (F_(1,12)_ < 1, *p* = 0.369), with a low R^2^ of 0.081. This indicates that MEP_test_/Mmax ratio cannot predict the SICI_MEPtest_. Then, a simple regression was calculated on mean values to predict the SICI_Mmax_ based on the MEP_test_/Mmax ratio (Figure 5D). A significant regression equation was found (F_(1,12)_ = 553.2, *p* < 0.001), with a large R^2^ of 0.980. Crucially, this indicates that the relationship between SICI and corticospinal excitability is relative to/dependent on the proportions of the spinal motoneurones pool activated by TMS, no matter how the corticospinal excitability is actively (levels of contraction) or passively (TMS intensity) increased.

## 4. Discussion

This study investigated the modulation of SICI when normalizing or not with the proportion of spinal motoneurones activated (SICI_Mmax_ vs. SICI_MEPtest_), with varying MEP_test_ amplitude. Our findings highlight the dependence of SICI measurement upon the proportion of motoneurones activated, i.e., SICI_Mmax_, regardless of the muscle state and TS intensity. The strong linear relationship between SICI_Mmax_ estimation and MEP_test_/Mmax ratio provides a relevant predictor of SICI for each individual. The main findings and their implications for future studies are discussed below.

### 4.1. Modulation of Corticospinal Excitability and SICI_MEPtest_


In the current study, we first measured the amplitude of unconditioned MEP (MEP_test_) by handling muscle state (at rest or at three levels of contraction) and TS intensity (120 or 130% of MT for each level of muscle contraction). As expected, corticospinal excitability increased with increasing muscle contraction and TS intensity [25,26].

When normalizing SICI with MEP_test_ (SICI_MEPtest_), we did not find any SICI_MEPtest_ modulation according to TS intensity. The current findings only revealed one significant difference between muscle states, with a reduction in SICI_MEPtest_ when tested at 30%MVIC and compared with rest. These results are therefore similar to those obtained in studies that evaluate SICI with no account of the proportion of motoneurones activated by the TS. Indeed, it is known that voluntary muscle contraction reduced SICI_MEPtest_ [16,27], particularly when tested with high levels of muscle contraction (>40% MVIC) [17,19]. One explanation is that during high levels of voluntary muscle contraction, the excitability of the spinal motoneurones pool is increased and the later indirect waves (called I3) generated by the TS (fixed to induce 1 mV MEP amplitude in the majority of previous studies) no longer contribute to the production of the MEP_test_ [28]. Indeed, the EMG response elicited by TS is usually composed of different waves, namely D-waves and I-waves. While direct D-wave is the first latency response originating from the activation of the pyramidal tract axons, the later indirect I-waves, numbering three waves (I1, I2, I3), are thought to result from the pyramidal neuron activation [29]. Because the CS predominantly inhibits the I3 [30], a decrease in SICI_MEPtest_ is observed at higher contraction levels (when there is no I3 to suppress) [18,27,31]. Finally, one has to keep in mind that a 1 mV fixed MEP amplitude across rest and active conditions (1) represents a different proportion of the spinal motoneurones pool activated by adjusted TMS intensity within an individual, and (2) does not result from the same contribution of descending waves activated by TMS, whereas SICI acts selectively on some of these descending waves.

Importantly, with low levels of muscle contraction [17,19] or different MEP_test_ amplitudes evoked at 120 or 130% of MT [11,32], the amount of SICI_MEPtest_ remains unchanged. As mentioned by Lackmy and Marchand-Pauvert (2010), this result seems logical from a purely mathematical point of view given the linear relationship between the MEP_cond_ and MEP_test_ amplitudes [9]. Therefore, the MEP_cond_-MEP_test_/MEP_test_ or MEP_cond_/MEP_test_ ratios result in a slope of an approximately linear relationship. This mathematical relationship explains why the SICI_MEPtest_ is not influenced by the MEP_test_ amplitude, whether the latter is increased by the level of muscle contraction or by TS intensity.

### 4.2. Modulation of SICI_Mmax_ and Its Relationship with MEP_test_/Mmax Ratio

By taking into account the proportion of motoneurones activated by the TS to quantify SICI, we found more inhibition with higher MEP_test_/Mmax ratio, both at rest and during the active muscle conditions. A major finding of the current study is the strong individual relationship between the SICI_Mmax_ estimation and the MEP_test_/Mmax ratios, which is ultimately independent from experimental conditions (muscle state and TS intensity). Increasing the level of muscle contraction or the TS intensity both led to a greater proportion of large spinal motoneurones activated by TMS and an increase in the MEP_test_ amplitude. Because of their higher threshold (when compared with smaller low-threshold motoneurones), these large motoneurones are more sensitive to a decrease in corticospinal input, induced by the CS [33]. By considering the recruitment order of motoneurones, it is possible that SICI_Mmax_ had a greater effect on the larger MEP_test_ amplitude and therefore the MEP_test_/Mmax ratio. Therefore, the greater the MEP_test_/Mmax ratio, the greater the SICI_Mmax_.

The individual relationship obtained makes it possible to individually predict SICI_Mmax_ from a specific MEP_test_/Mmax ratio. This result is of particular methodological importance given the crucial contribution of SICI modulations to motor skill acquisition [34,35] or acute and multi-session strength training programs [3,36,37]. Several recent articles highlighted the high heterogeneity in the setting parameters, such as muscle state, used to evaluate SICI modulation before and after motor practice [2,17,38,39]. Some authors argued that, when the muscle is active, measurements of intracortical and corticospinal excitability are more sensitive and specific to motor learning than the same tests performed at rest after motor learning/training [40]. This statement, however, does not indicate the level(s) of muscle contraction to choose during testing. The findings of the current study are also interesting for protocols assessing SICI changes following an acute fatiguing exercise [40]. In these protocols, post-test measures must be performed immediately following the exercise in order to avoid a recovery influence. Fatigue is known to increase the size of MEP_test_ [41,42]. Therefore, adjusting TS intensity at the beginning of post-test SICI measurement to obtain a similar MEP_test_ amplitude than during pre-test (such as 1 mV) is a time-consuming procedure that could bias the interpretation of fatigue-induced modulation of SICI.

It should be noted that while we observed a greater amount of SICI_Mmax_ with a greater MEP_test_/Mmax ratio, Lackmy and Marchand-Pauvert (2010) reported a decrease in SICI_Mmax_ when the MEP_test_/Mmax ratio was higher than 30%. In their study, TMS-evoked responses were recorded in small hand muscles with increasing TS intensities for a given CS. The authors found a peak of inhibition when the MEP_test_/Mmax ratio was approximately equal to 20–30%, explained by a greater inhibitory effect of SICI on large and high-threshold motoneurones (i.e., larger MEP_test_/Mmax ratio). This peak of inhibition was then followed by a decrease in the SICI_Mmax_. When increasing TS intensities, and therefore MEP_test_/Mmax ratios, greater short-latency direct-waves (D-waves) are recruited [43]. It is known that these D-waves are less sensitive to SICI [8,14,30], resulting in SICI_Mmax_ decrease with increasing MEP_test_/Mmax ratios. Concurrently, in this study, SICI_Mmax_, MEP_test_ and Mmax were tested in a more proximal muscular group (i.e., forearm muscle) than the above-cited study. This choice of testing FCR muscle was mainly motivated by the common use of wrist muscles in numerous strength-training programs that require pre-intervention and post-intervention SICI measurements [37,38,44,45,46]. The recruitment and discharge properties of motoneurones are known to be different between small hand muscles compared with more proximal arm muscles [47]. During voluntary contractions, MEPs amplitude evoked in a hand muscle at a fixed TS intensity was almost saturated at 20%MVIC whereas MEPs amplitude evoked at the same intensity in the FCR reached a plateau at 40%MVIC [47]. These characteristics probably explain why, in the current study, we did not observe a decrease in SICI_Mmax_ with MEP_test_/Mmax ratio greater than 30–40%.

### 4.3. Methodological Recommendations

By calculating the individual slopes, intercept, R^2^ and *p*-values between the MEP_test_/Mmax ratio and the SICI_Mmax_ before an intervention (as carried out in the present study), it becomes possible to overcome the testing of a specific level of muscle contraction, or to perform a size MEP_test_ adjustment in order to test SICI_Mmax_ modulation after motor practice in healthy subjects. It only requires the calculation of the mean SICI_Mmax_ for a specific MEP_test_/Mmax ratio after any intervention. Then, it is possible to compare this SICI_Mmax_ value with the equation of the regression obtained before the intervention in order to evaluate the SICI_Mmax_ modulation induced. There are two possible results: (1) the value obtained in post-tests is aligned with the theoretical linear relationship found in pre-tests, indicating no change in SICI_Mmax_ following the intervention; and (2) the value obtained in post-tests is above/under the theoretical line, indicating that for a given MEP_test_/Mmax ratio, SICI_Mmax_ decreases/increases following the intervention. Importantly, in the current study, the individual regression analyses are based on eight data points per individual (i.e., 2 TS intensities and four levels of muscle contractions). A quicker approach can be considered, with a more restricted number of points. For example, individual regression analyses based on only four points, corresponding to different levels of muscle contractions with a unique TS intensity, also lead to strong individual relationships between the SICI_Mmax_ estimation and the MEP_test_/Mmax ratios (see Appendix A). It is important to keep in mind that these methodological recommendations are based on results obtained in a study completed acutely. Since Mmax does not always remain constant during the course of an experiment [48] and has been shown to be altered following strength-training [49], it is crucial to repeat the testing for each experimental condition, including during post-test measurements.

### 4.4. Limitations

The current study presents certain limitations that should be noted. First, the number of pulses selected to evaluate SICI is quite smaller than the one recommended for highly reliable SICI estimates (i.e., more than 25 pulses) [50]. This choice was made to avoid any fatigue effect that might have potentially confounded the results obtained. Moreover, both SICI_MEPTEST_ and SICI_Mmax_ modulations also depend on CS intensity and interstimulus interval (ISI) duration [6,12,51,52,53], two fixed parameters whose effects were not assessed in the current study. The CS intensity was fixed at 70% of MT calculated according to each levels of contraction. This was done based on a previous study showing that during active condition, the higher level of inhibition was obtained delivering a CS of 70% aMT [16]. Of note, a recent study has demonstrated that the CS intensity (60%, 70% or 80% aMT) did not alter the modulation of SICI_MEPTest_ assessed at different muscle contraction levels [17]_._ Concerning the ISI, it is known that SICI_MEPTEST_ consists of two phases of inhibition according to the ISI duration [6,12]. The first one is observed at ISI = 1 ms and is thought to reflect the refractoriness of axons of cortical interneurons subliminally activated by a subthreshold CS [6], and/or extrasynaptic GABA tone [54]. The second one observed at ISI = 2.5/3 ms accounted for synaptic inhibition mediated by the GABA_A_ receptors [6,12,55]. The 3-ms ISI was selected in the current study as this duration consistently demonstrates a clear peak of inhibition [53,56]. However, future studies should address whether CS intensity and ISI duration can affect SICI_Mmax_ modulation. Finally, only one forearm muscle was tested, and we did not consider (1) antagonist muscle and (2) synergistic muscles that are also relevant to force production without a concomitant increase in the EMG of the FCR agonist muscle. At this stage, the results cannot be generalized to all muscle groups and deserve further investigations.

## 5. Conclusions

The current study provides new methodological insights into how SICI, representing an inhibitory phenomenon known to act purely at the cortical level, is ultimately dependent upon the proportion of motoneurones activated by TMS. Moreover, based on the results obtained, our study suggests a new way to evaluate SICI_Mmax_ modulation that makes it possible to get away from important methodological constraints related to pre/post intervention measures. This may have practical implications for the assessment of cortical adaptations that contribute to functional changes induced by motor practice, motor learning, motor exercise or strength training in future studies.

## Figures and Tables

**Figure 1 brainsci-10-00433-f001:**
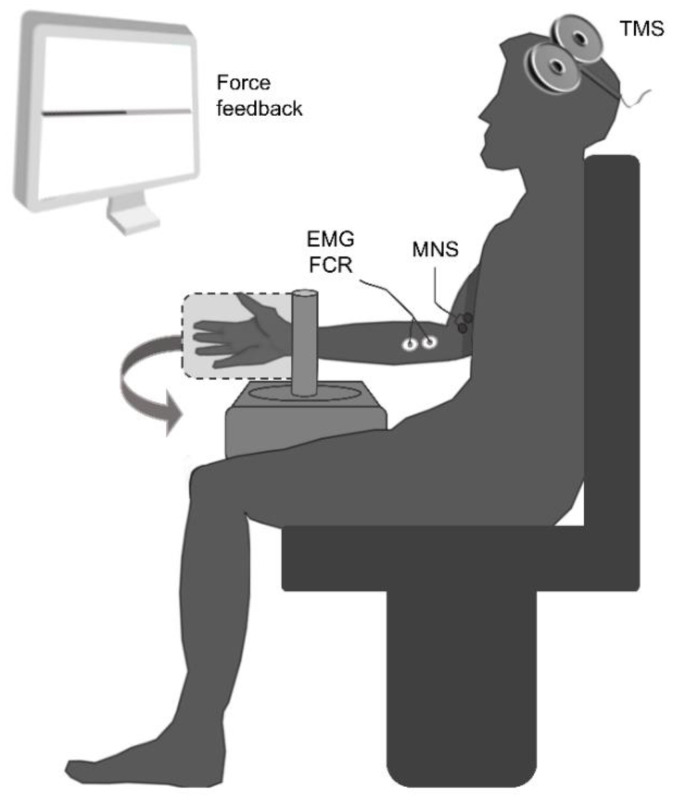
Schematic view of the experimental setup. Transcranial magnetic stimulation (TMS); median nerve stimulation (MNS); electromyographic (EMG); flexor carpi radialis (FCR).

**Figure 2 brainsci-10-00433-f002:**
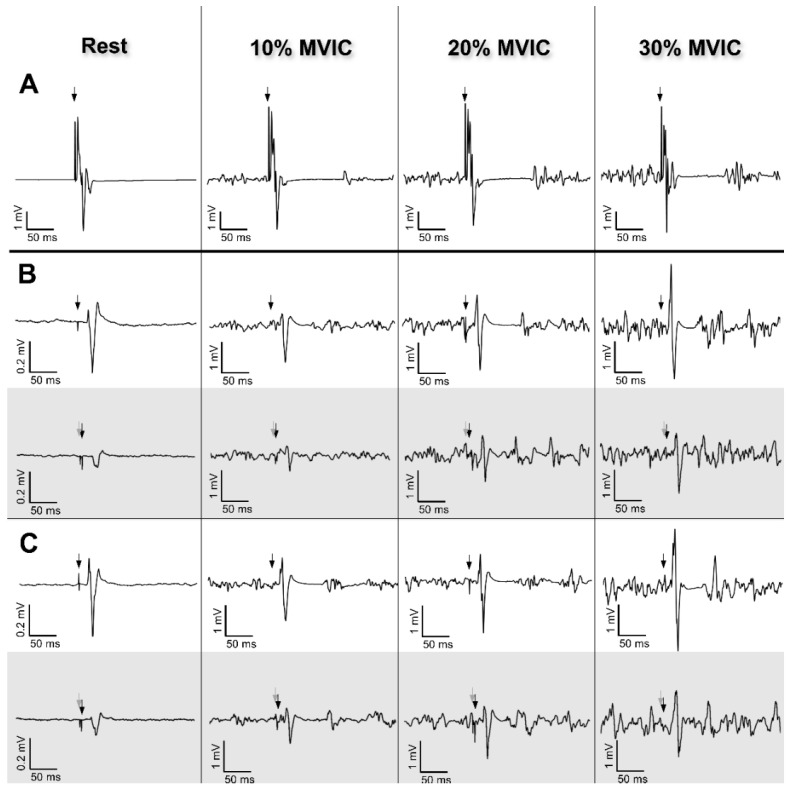
Typical raw data recordings of the right FCR muscle depicted at rest, 10%, 20% and 30% of maximal voluntary isometric contraction (MVIC) for a representative participant. Black and grey vertical arrows indicate time of stimulation for test and conditioning stimuli, respectively. (**A**) Maximal M-wave (Mmax) induced by electrical stimulation. (**B**) Unconditioned (white) and conditioned (grey) motor-evoked potentials (MEPs) with the test stimulus induced at 120% of the motor threshold. (**C**) Unconditioned (white) and conditioned (grey) MEP with the test stimulus induced at 130% of the motor threshold. Note that the *y*-axis scale for MEPs amplitude differs between resting and active conditions (i.e., 0.2 mV vs. 1 mV) for a better graphical visualization.

**Figure 3 brainsci-10-00433-f003:**
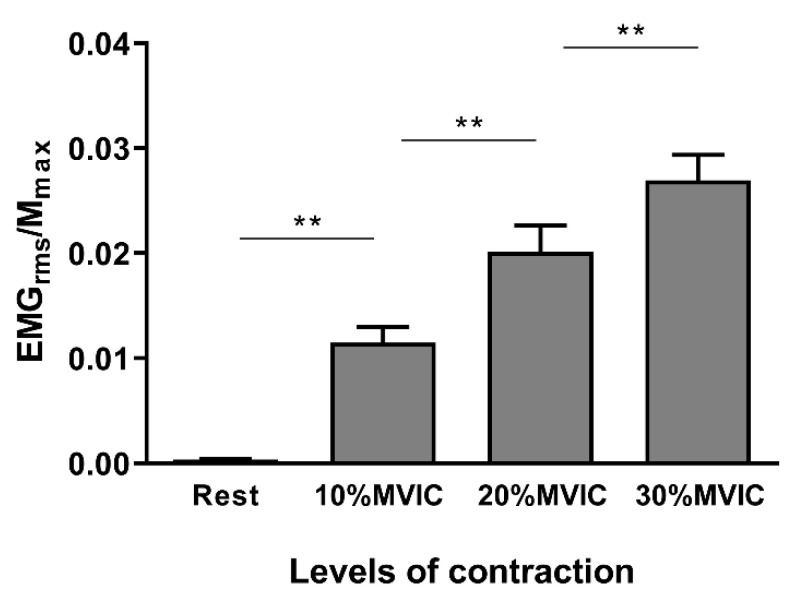
EMG_rms_/Mmax ratios pooled for MEP_test_ (unconditioned) and MEP_cond_ (conditioned) and the two test stimulus (TS) intensities (120% and 130% of motor threshold (MT)). EMG_rms_ were recorded 100 ms before the delivery of each TMS pulse. ** *p* < 0.01.

**Figure 4 brainsci-10-00433-f004:**
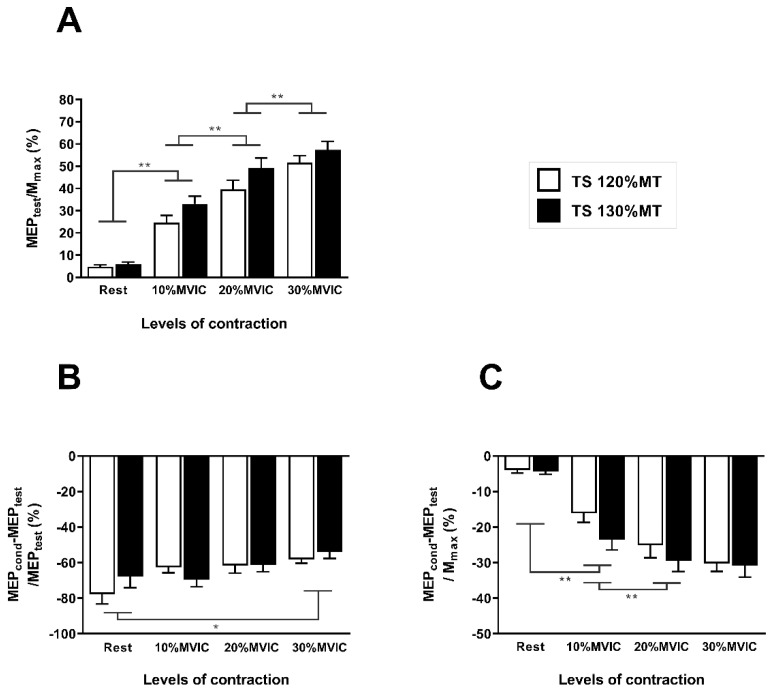
(**A**) MEP_test_/Mmax ratios according to the TS intensity and the levels of contraction. (**B**) Short-interval intracortical inhibition (SICI) expressed as percentage of MEP_test_ (SICI_MEPtest_) according to the TS intensity and the levels of contraction. (**C**) SICI expressed in percentage of Mmax (SICI_Mmax_) according to the TS intensity and the levels of contraction. * *p* < 0.05; ** *p* < 0.001.

**Figure 5 brainsci-10-00433-f005:**
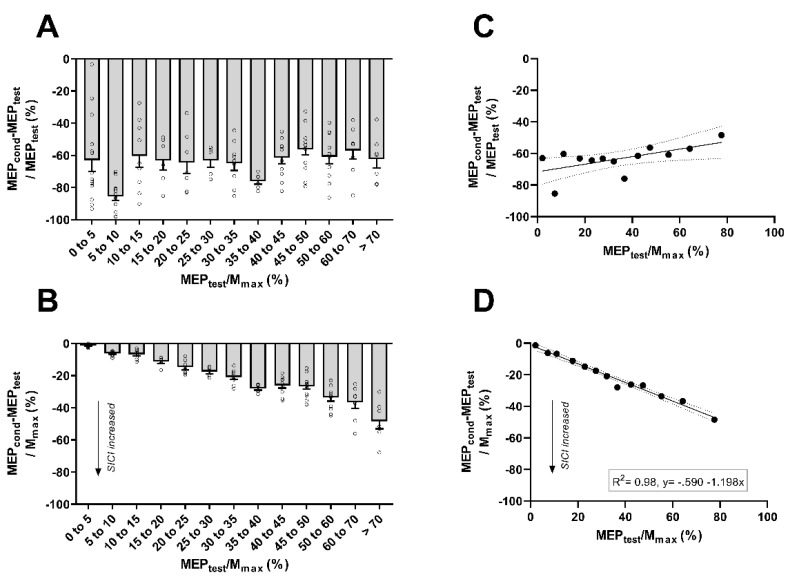
SICI normalized to MEP_test_ (SICI_MEPtest_) (**A**) and SICI normalized to Mmax (SICI_Mmax_) (**B**) organized into different bins based on the MEP_test_/Mmax ratio. Grey bars represent the mean value for each bin, with standard error of the mean. Black dots represent individual values included in each bin. Non-significant regression plots between MEP_test_/Mmax ratio SICI_MEPtest_ (**C**) and significant regression plots between MEP_test_/Mmax ratio SICI_Mmax_ (**D**) Black dots in regressions represent the mean value for each bin. Grey lines in the regression figures represent 95% confident interval.

**Table 1 brainsci-10-00433-t001:** Comparison of motor thresholds (MT) (%maximum stimulator output (MSO)), single- and paired-pulse TMS raw EMG_rms_ for MEP data (μV), EMG_rms_ expressed as a% of EMG_rms_ obtained during MVIC; Mmax amplitude (mV) and raw EMG_rms_ for Mmax data (μV) across levels of muscle contraction (%MVIC). Data represent mean ± SD. The MT for the Rest condition was significantly higher than all the MT obtained for active conditions (10, 20 and 30% MVIC).

Variables	Levels of Muscle Contraction (%MVIC)			
	Rest	10%	20%	30%
MT (%MSO)	40.87 ± 7.3 **	31.73 ± 8.1	31.53 ± 7.2	31.27 ± 7
EMGrms MEP (μV)	1.97 ± 0.8	67.18 ± 28.2	113.58 ± 46.5	160.32 ± 64.5
EMGrms (%EMGrms-MVIC)	0.31 ± 0.2	9.65 ± 5.7	16.06 ± 8.7	21.96 ± 10.5
Mmax (mV)	6.97 ± 3.7	6.82 ± 3.8	6.60 ± 3.7	6.70 ± 3.9
EMGrms Mmax (μV)	1.75 ± 0.6	65.04 ± 27.7	114.83 ± 61	160.73 ± 78.8

** *p* < 0.001. motor thresholds (MT); maximum stimulator output (MSO); root mean square (rms); electromyographic (EMG); motor-evoked potential (MEP); maximum voluntary isometric contraction (MVIC).

**Table 2 brainsci-10-00433-t002:** Regression analysis between MEP_test_/Mmax ratio (independent variable) and SICI_Mmax_ (dependent variable) estimated individually for each subject. Individual slopes, intercept, R^2^, *p* values and range of the MEP/Mmax ratios were extracted and reported.

Subject	Slope	Intercept	R^2^	*p* Value	Range of the MEP_test_/Mmax Ratios (%)
1	−0.553	0.118	0.911	<0.001	1.19–46.9
2	−0.798	0.508	0.944	<0.001	2.97–32.91
3	−0.593	−0.320	0.879	=0.001	7.54–46.85
4	−0.546	−1.474	0.895	<0.001	1.23–46.74
5	−0.518	−1.143	0.806	=0.002	6.40–45.38
6	−0.693	0.728	0.941	<0.001	1.11–47.68
7	−0.725	−1.884	0.943	<0.001	6.07–72.37
8	−0.455	−6.172	0.848	=0.001	10.66–74.25
9	−0.626	−1.282	0.898	<0.001	2.44–69.65
10	−0.486	−3.472	0.947	<0.001	8.28–62.33
11	−0.610	0.231	0.961	<0.001	1.52–58.23
12	−0.466	−2.872	0.856	=0.001	5.38–49.67
13	−0.405	−5.942	0.791	=0.003	7.29–66.39
14	−0.385	−1.949	0.921	<0.001	1.75–79.66
15	−0.671	−1.563	0.937	<0.001	5.84–89.71
Mean	−0.569	−1.706	0.898		4.65–59.25
SD	0.120	2	0.053		3.09–15.78

## Data Availability

https://osf.io/jhaus/.

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
