# Peer review of "Influence of Voluntary Contraction Level, Test Stimulus Intensity and Normalization Procedures on the Evaluation of Short-Interval Intracortical Inhibition"

_brainsci, 2020, doi:10.3390/brainsci10070433_

Round 1

Reviewer 1 Report

Starting from the work by Lackmy & Marchand-Pauvert (2010), and by Opie & Semmler (2014), the authors tested the effects of different test stimulus intensities and of different levels of muscle contractions on SICI, normalizing the results to MEP test (SICI MEPTEST) and to Mmax, i.e. the maximal compound muscle action potential, (SICI Mmax).

The novelty of the work is partial, since the work of Lackmy et al. 2010 already suggested that SICI is less variable (inter- and intra- subjects variability) when the data are normalized to Mmax, and the work by Opie & Semmler 2014 already showed that SICI normalized to Mmax increases with slight muscle activation. Here the authors tested different levels of muscle activation to gain further insight on the proportion of motoneurones activated by the TMS and studied through Mmax. They also performed a regression analysis showing that SICIMmax can be predicted in individual subjects from a specific MEPtest/Mmax ratio, while MEPtest/Mmax ratio could not predict the SICIMEPtest.

The study is well designed and data analysis seems well performed. Conversely, the writing of the paper should be improved, in particular in the introduction (the literature background should be more properly presented) and the discussion (discussion should be more clear).

Specific points:

Introduction

- Line 33-34: “the reduction of intracortical inhibition within M1 is a crucial part of neural adaptation following acute and multi-session strength training programs[2,3]”. Reference #3 does not support this statement (the conclusion of the meta-analysis is that SICI is NOT reduced after a single bout of strength training).

- Line 37-38: “Short-interval intracortical inhibition (SICI) is the most widely used paired-pulse measure to evaluate inhibitory circuits”. Please provide a reference or rephrase

- line 45: “The amount of SICI is conventionally quantified by calculating the MEPcond/MEPtest ratio or the difference between the mean MEPcond and the mean MEPtest then expressed as a percentage of the mean MEPtest amplitude (SICIMEPtest) [6]”. In reference #6 (Kujirai et al. 1993), the size of the conditioned responses is expressed as a percentage of the size of the control response and this is the usual way SICI is expressed. Please provide the references for the other quantification methods

- previous literature about difference in SICI in agonist and antagonist muscles could be included

Materials and Methods

- please specify if the healthy volunteers had a negative neurological and psychiatric history and if they were taking any psychoactive drug

- I suggest adding a figure about the experimental setup (to clarify subjects’ position during the active task and the setup of the peripheral stimulation)

- please clarify if single pulses were interspersed with paired pulses or performed in different recording blocks

- please specify the interstimulus interval between single pulses and which was the mean duration of the recordings during active contraction

- was the >95% confidence interval amplitude criterium applied for AMT to all the subjects or only to the 4 participants with MEPs> 200uV?

- figure 1A suggests that stimulation artifacts were very close to muscle action potentials. How was this potential confounder managed during data analysis? Please clarify

- was the 70%AMT pulse sufficiently low to be subthreshold also during the 30% maximal voluntary contraction blocks?

Discussion

- please clarify the choice of testing flexor carpi radialis muscle

- line 311-312 “decrease in SICIMEPtest is observed at higher contraction levels (when there is no I3 to suppress) “. please also discuss any possible effect of SICF on SICI reduction during muscle activation (reference Ortu et al 2008 J. Physiol. 586, 5147–5159)

- did baseline MEP amplitude predict SICI, as in many literature papers on the topic? Please discuss

- figure 2A and figure 2D look specular, while SICIMEPtest in figure 2C has almost an opposite behavior compared to SICIMmax in figure 2D. Please better discuss how any mathematical relationship derived from the formulas can explain this very different findings about the same phenomenon, and please better clarify why these differences are not purely mathematical

Author Response

We would like to thank the Reviewers for taking the time to review our manuscript. We believe the comments and suggestions received will significantly strengthen the article. Hence, we have endeavored to address each comment as both a point-by-point response below, and within the updated version of the manuscript. In addition, a native English speaker edited our revised manuscript. 

Starting from the work by Lackmy & Marchand-Pauvert (2010), and by Opie & Semmler (2014), the authors tested the effects of different test stimulus intensities and of different levels of muscle contractions on SICI, normalizing the results to MEP test (SICI MEPTEST) and to Mmax, i.e. the maximal compound muscle action potential, (SICI Mmax).

The novelty of the work is partial, since the work of Lackmy et al. 2010 already suggested that SICI is less variable (inter- and intra- subjects variability) when the data are normalized to Mmax, and the work by Opie & Semmler 2014 already showed that SICI normalized to Mmax increases with slight muscle activation. Here the authors tested different levels of muscle activation to gain further insight on the proportion of motoneurones activated by the TMS and studied through Mmax. They also performed a regression analysis showing that SICIMmax can be predicted in individual subjects from a specific MEPtest/Mmax ratio, while MEPtest/Mmax ratio could not predict the SICIMEPtest.

The study is well designed and data analysis seems well performed. Conversely, the writing of the paper should be improved, in particular in the introduction (the literature background should be more properly presented) and the discussion (discussion should be more clear).

We would like to thank the Reviewer#1 for the valuable comments and suggestions, which we believe have improved the quality of the paper. We hope that the reviewer will find our responses to the comments satisfactory.

Specific points:

Introduction

- Line 33-34: “the reduction of intracortical inhibition within M1 is a crucial part of neural adaptation following acute and multi-session strength training programs[2,3]”. Reference #3 does not support this statement (the conclusion of the meta-analysis is that SICI is NOT reduced after a single bout of strength training).
RESPONSE: The problematic reference has been removed and we have also added the first study to show that SICI was reduced following strength training “Strength Training Reduces Intracortical Inhibition” by Weier, Pearce and Kidgell (2012).

- Line 37-38: “Short-interval intracortical inhibition (SICI) is the most widely used paired-pulse measure to evaluate inhibitory circuits”. Please provide a reference or rephrase
RESPONSE: The sentence has been modified :

Line 41-42: “Short-interval intracortical inhibition (SICI) is a well-established paired-pulse measure to evaluate inhibitory circuits within the M1 area.”

- line 45: “The amount of SICI is conventionally quantified by calculating the MEPcond/MEPtest ratio or the difference between the mean MEPcond and the mean MEPtest then expressed as a percentage of the mean MEPtest amplitude (SICIMEPtest) [6]”. In reference #6 (Kujirai et al. 1993), the size of the conditioned responses is expressed as a percentage of the size of the control response and this is the usual way SICI is expressed. Please provide the references for the other quantification methods
RESPONSE: Following Reviewer’s concern, the conventional SICI quantification methods has been clarified and a reference has been added for the other quantification methods :   

Line 50-51: The amount of SICI is conventionally quantified by expressing the mean MEPcond amplitude as a percentage of the mean MEPtest amplitude [5] or the difference between the mean MEPcond and the mean MEPtest then expressed as a percentage of the mean MEPtest amplitude (SICIMEPtest) [9].”

- previous literature about difference in SICI in agonist and antagonist muscles could be included
RESPONSE: We agree with Reviewer #1 that SICI modulation can vary in agonist or antagonist muscle, in particular following strength-training program (Mason et al. 2019), but the aim of this methodological paper was not to discuss about this specific SICI modulation.
However, we have acknowledged in the discussion section that we did not assess the antagonist muscle EMG in our study, which constitutes a limit.

Line 451-454: “Finally, only one forearm muscle was tested and we did not consider 1) antagonist muscle and 2) synergistic muscles that are also relevant for force production without a concomitant increase in the EMG recorded in the FCR agonist muscle.”

Materials and Methods

- please specify if the healthy volunteers had a negative neurological and psychiatric history and if they were taking any psychoactive drug
RESPONSE: Following Reviewer’s suggestion, this has been specified:

Line 107-108: “All volunteers were screened by a medical doctor for contraindications to TMS [21] and were excluded if they a had history of psychiatric or neurological illness.”

- I suggest adding a figure about the experimental setup (to clarify subjects’ position during the active task and the setup of the peripheral stimulation)
RESPONSE: We have added a Figure (Figure 1) about the experimental setup.

Figure 1: Schematic view of the experimental setup. TMS = transcranial magnetic stimulation; MNS = median nerve stimulation; EMG = electromyographic; FCR = flexor carpi radialis.

- please clarify if single pulses were interspersed with paired pulses or performed in different recording blocks
RESPONSE: We agree with Reviewer#1 that this information was missing and we have added the following sentence:

Line 130-132: “Single-pulse and paired-pulse TMS measures obtained with the two TS intensities were performed in different recording blocks and the order of the blocks was randomized order across participants.”

- please specify the interstimulus interval between single pulses and which was the mean duration of the recordings during active contraction
RESPONSE: This has been specified:

Line 132: “The inter-trial intervals between single-pulses was at least 3 s [22].”

Line 132-134: “During active conditions, the mean duration of the recording was set at 500 ms and participants were asked to relax after the stimulation for few seconds before the next trial and contracting again.”

- was the >95% confidence interval amplitude criterium applied for AMT to all the subjects or only to the 4 participants with MEPs> 200uV?
RESPONSE: This criterion was applied only to the 4 participants with MEPs > to 200 mV.
To make this choice, we both considered 1) the conventional aMT definition adopted in a large majority of previous studies regardless of the level of contraction (i.e. the lowest stimulus intensity (% MSO) required to produce MEPs greater than 200 μV in at least 5 out of 10 consecutive stimuli.)  2) the findings of Ortu et al. (2008) who assessed SICIMEPTest at various levels of contraction, with a high number of methodological controls. They stated that “For 25% and 50% MVC, the AMT (AMT25% and AMT50%, respectively) was defined as the minimum stimulus intensity that produced at least five MEPs from 10 consecutive trials with a peak-to-peak amplitude greater than the 95% confidence interval of the prestimulus mean EMG activity (Mills & Nithi, 1997)” and we followed exactly the same procedure in our study.  

- figure 1A suggests that stimulation artifacts were very close to muscle action potentials. How was this potential confounder managed during data analysis? Please clarify
REPONSE: Mmax mean latency (≈3.5 ms) is effectively shorter than mean MEPs latency evoked in FCR muscle. However, the data analysis was performed carefully and manually. By zooming on the signal, the Mmax peak-to-peak amplitude was clearly identified after the stimulation artifact (see below raw traces in the software used to analyze data, for a participant with a representative mean Mmax amplitude).
Note that the EMG signal returns to the baseline level after the stimulation artifact (green arrow) which makes it possible to clearly distinguish the beginning of the M-wave.

- was the 70%AMT pulse sufficiently low to be subthreshold also during the 30% maximal voluntary contraction blocks?
REPONSE: We have calculated the aMT for each level of contraction. Then, for the 30% of MVIC condition, the 70 %aMT corresponds to 70 % of the aMT30%MVIC (i.e. assessed during the same level of muscle contraction), to ensure that the CS was sufficiently low to be subthreshold.
Of note, we found that the aMT did not significantly differ depending on the level of muscle contraction (line 242).

Discussion

- please clarify the choice of testing flexor carpi radialis muscle
RESPONSE: Following Reviewer’s concern, we have added the following sentence to justify the choice of testing FCR.

Line 440-442: “This choice of tested FCR muscle was mainly motivated by the common use of wrist muscles in numerous strength-training programs that require pre-intervention and post-intervention SICI measurements [40,41,47–49].” 

Moreover, if in future studies SICI measures are a part of a more global project in which other neurophysiological measurements will be performed, such as H-reflex recording, FCR muscle is more appropriate than FDI muscle (where H-reflex at rest is almost impossible to evoke).

- line 311-312 “decrease in SICIMEPtest is observed at higher contraction levels (when there is no I3 to suppress) “. please also discuss any possible effect of SICF on SICI reduction during muscle activation (reference Ortu et al 2008 J. Physiol. 586, 5147–5159)
RESPONSE: We agree with Reviewer#1 that this could have been a possible effect for SICI measured during low contraction levels.
However, Ortu et al. (2008) suggested that SICI may be “contaminated” with concurrently superimposed recruitment of high-threshold excitatory interneurons when the CS intensity was set at ≥80 % aMT : “At low contraction levels, SICI with S1 ≥ 80% AMT is reduced compared to rest by a superimposed recruitment of SICF”. (Ortu et al. 2008).
In the current study, the CS Intensity was set at 70% of aMT. This low-intensity stimulus allowed us to test SICI independently of the effects on SICF at low contraction levels: “Low intensity (70%AMT) conditioning stimuli can test SICI independently of effects on SICF at low contraction levels”; “At these intensities, SICI is unaffected by weak voluntary contraction” (Ortu et al. 2008).
The results obtained are congruent with Ortu et al. (2008) since we found no change in SICIMEPtest assessed during 10% MVIC when compared to rest, with a CS set at 70%aMT.
Finally, a possible effect of SICF on SICI reduction during muscle activation does not seem possible for higher level of muscle contraction as suggested by Ortu et al. (2008): “the recruitment of SICF is facilitated during slight voluntary contraction (10% MVC), but less so or not at all at 25% and 50% MVC.”

- did baseline MEP amplitude predict SICI, as in many literature papers on the topic? Please discuss
RESPONSE: Following Reviewer’s suggestion, we have calculated correlations between baseline raw MEPtest amplitude (rest condition) and SICIMEPTest for both TS 120 %rMT (left panel) and TS 130 %rMT (right panel).
We did not find significant correlations between MEPtest amplitude and SICIMEPTest with TS 120 %rMT (R2 = 0.146; p = 0.159) or between MEPtest amplitude and SICIMEPTest with TS 130 %rMT (R2 = 0.130; p = 0.186). This may be due to a ‘ceiling effect’ for SICIMEPtest evoked in many previous papers (i.e. when inhibition approaches 100%) [Cirillo et al., 2018; Sanger et al., 2001]. SICIMEPtest phenomenon is also subject to noise from raw MEP amplitude variability [Kiers et al., 1993] and probably prevent, in the current study, any correlation to be observed.

However, when MEPtest amplitudes were normalized to Mmax (MEPtest/Mmax (%)) and SICI was normalized to Mmax (MEPcond-MEPtest/Mmax (%)), a normalization that takes into account the proportions of the spinal motoneurones pool activated by TMS, we found a significant regression (F(1,12) = 553.2, p < .001), with a large R2 of 0.980 (see Figure 5D). This indicates that it is possible to individually predict SICIMmax from a specific MEPtest/Mmax ratio, with a higher amount of SICIMmax for a greater MEPtest/Mmax ratio. Crucially, this indicates that the relationship between SICI and corticospinal excitability is relative to the proportions of the spinal motoneurones pool activated by TMS.

- figure 2A and figure 2D look specular, while SICIMEPtest in figure 2C has almost an opposite behavior compared to SICIMmax in figure 2D. Please better discuss how any mathematical relationship derived from the formulas can explain this very different findings about the same phenomenon, and please better clarify why these differences are not purely mathematical
RESPONSE: “figure 2A and figure 2D look specular”: since figure 2A initially depicted EMGrms values, we believed that the Reviewer refers to Figure 2B and 2D ?

Reviewer 2 Report

In the study by Neige et al., the authors investigated whether the responses of short-intracortical inhibition (SICI) are dependent upon the proportion of motoneurons activated by TMS. The authors tested this by varying the intensity of the stimulation (MEPtest) and muscle activity (four varying levels of intensity) conditions and comparing the responses of SICI that either take into account spinal activity (Mmax) or not (SICI as a function of MEPtest values). While the results of this study is convincing for the parameters selected by the authors, there are other factors that are important to consider when doing this comparison, such as the conditioning stimulus intensity. Moreover, the article contains many run-on sentences that makes the text hard to read at times. I recommend that the authors consult a native English speaker to edit the text in a more concise manner. Below are some concerns I have with the manuscript:

One of the issues I have with the article relates to how the aim of the study is motivated within the introduction. There are many nomenclatures and details that are discussed that makes the text hard to follow at times. I would suggest limiting this as much as possible and focus on how the study is testing something different from previous investigations. In other words, how different is this from Opie and Semmler 2014 and why are the additional conditions tested in this manuscript important to consider?

Beyond the way in which SICI is defined (i.e. as an expression of mean MEPtest or Mmax), other parameters  such as the conditioning stimulus (CS) intensity and inter-stimulus intervals (ISI) are important to consider as they can importantly produce a range of affects towards the MEP response (See Ibanez et al.: https://doi.org/10.1016/j.brs.2019.11.002). The authors should introduce this topic and discuss why they have selected 70% CS and ISI of 3ms. Along these lines, would you presume measuring SICI-Mmax with ISI of 1ms to be effective as the mechanism underlying this response is though to be dramatically different from the SICI recorded at 3ms ISI? This could be added as a discussion point.

The introduction will benefit with the logic as to why Opie and Semmler 2014 tested SICI in an active condition. In other words, measuring SICI-Meptest during activation is well-known to reduce the inhibitory effect. Should the prediction be the same for SICI-Mmax?

Ortu et al;https://doi.org/10.1113/jphysiol.2008.158956 found that muscle activity effects on SICI were stronger when the CS intensity was larger (80 and 90% rMT). How do the results seen here take into account the differences found across CS intensity found in Ortu et al? Alternatively, could it be that 70% CS is only marginally effective at suppressing I-waves regardless of the condition in which it is probed? It would be important test whether a more effective CS intensity (either 80 or 90%) on the MEP would also show the same/different response seen in the SICI-Mmax at 90% CS.

Author Response

We would like to thank the Reviewers for taking the time to review our manuscript. We believe the comments and suggestions received will significantly strengthen the article. Hence, we have endeavored to address each comment as both a point-by-point response below, and within the updated version of the manuscript. In addition, a native English speaker edited our revised manuscript. 

In the study by Neige et al., the authors investigated whether the responses of short-intracortical inhibition (SICI) are dependent upon the proportion of motoneurons activated by TMS. The authors tested this by varying the intensity of the stimulation (MEPtest) and muscle activity (four varying levels of intensity) conditions and comparing the responses of SICI that either take into account spinal activity (Mmax) or not (SICI as a function of MEPtest values). While the results of this study is convincing for the parameters selected by the authors, there are other factors that are important to consider when doing this comparison, such as the conditioning stimulus intensity. Moreover, the article contains many run-on sentences that makes the text hard to read at times. I recommend that the authors consult a native English speaker to edit the text in a more concise manner. Below are some concerns I have with the manuscript:
Thank you for the useful comments and suggestions, we have made a number of revisions in accord with your points (see below).

One of the issues I have with the article relates to how the aim of the study is motivated within the introduction. There are many nomenclatures and details that are discussed that makes the text hard to follow at times. I would suggest limiting this as much as possible and focus on how the study is testing something different from previous investigations. In other words, how different is this from Opie and Semmler 2014 and why are the additional conditions tested in this manuscript important to consider?
RESPONSE: The current study differs on several points from Opie & Semmler (2014) and we have shortened and changed the introduction in depth to highlight these differences for the reader (see the modifications in the main test as well as the Response below concerning the Introduction).

First, there was only one level of contraction tested in the study of Opie & Semmler (2014) (i.e. 5 %MVIC). SICIMEPtest has been shown to decrease according to the specific level of contraction (40% of MVIC when compared to 5% and 20% of MVIC [Rantalainen et al., 2013]). Since many studies analyze SICI phenomenon for force level > 5% of MVIC, we believed that the assessment of several contraction levels was of importance in regards to SICIMmax normalization. Moreover, Opie & Semmler (2014) focused their study on the effect of the test stimulus intensity but did not mention a correlation between SICI and MEPtest/Mmax ratio.
The originality of our study is to demonstrate the existence of a strong relationship between SICIMmax estimation and MEPtest/Mmax ratio.
Finally, Opie & Semmler (2014) tested the FDI muscle whereas we selected the FCR muscle due to the common use of wrist flexors in numerous strength-training programs which require pre-intervention and post-intervention SICI measurements [Hendy and Kidgell, 2013; Mason et al., 2019b; Mason et al., 2019a; Nuzzo et al., 2017].

Beyond the way in which SICI is defined (i.e. as an expression of mean MEPtest or Mmax), other parameters  such as the conditioning stimulus (CS) intensity and inter-stimulus intervals (ISI) are important to consider as they can importantly produce a range of affects towards the MEP response (See Ibanez et al.: https://doi.org/10.1016/j.brs.2019.11.002). The authors should introduce this topic and discuss why they have selected 70% CS and ISI of 3ms. Along these lines, would you presume measuring SICI-Mmax with ISI of 1ms to be effective as the mechanism underlying this response is though to be dramatically different from the SICI recorded at 3ms ISI? This could be added as a discussion point.
RESPONSE: We agree with Reviewer #1 that the CS intensity and the ISI are two crucial parameters known to affect SICI modulation.
Concerning the ISI duration, two phases of SICI has been identified underlying distinct physiological mechanisms (i.e. ISI of 1 ms and ISI of 2.5/3 ms). The first phase (ISI 1 ms) may be explained by refractoriness of axons of cortical interneurons subliminally activated by a subthreshold CS (Fisher et al. 2002) but also by extrasynaptic GABA tone (Stagg et al. 2011; Roshan et al. 2003). The second phase (ISI of 2.5/3 ms) is mediated by synaptic mechanisms (Roshan et al. 2003). At this point, it would be difficult to infer directly about SICIMmax with ISI of 1 ms since the underlying mechanisms are different according to the tested ISI. We will keep this thought in mind for future studies.

Following the Reviewer’s concern, the following paragraph has been added in the discussion (limitations section):

Line 438-451: “Moreover, both SICIMEPTest and SICIMmax modulations also depend on CS intensity and ISI duration [Fisher et al., 2002; Ibáñez et al., 2020; Neige et al., 2020; Roshan et al., 2003; Vucic et al., 2009], two fixed parameters whose effects were not assessed in the current study. The CS intensity was fixed at 70% of the MT calculated according to each levels of contractions. This was done based on a previous study showing that during active condition, the higher level of inhibition was obtained delivering a CS of 70% aMT [Ortu et al., 2008]. Of note, a recent study has demonstrated that the CS Intensity (60%, 70% or 80% aMT) did not alter the modulation of SICIMEPTest assessed at different muscle contraction levels [Hendy et al., 2019b]. Concerning the ISI, it is known that SICIMEPTEST consists of two phases of inhibition according to the ISI duration [Fisher et al., 2002; Roshan et al., 2003]. The first one is observed at ISI=1ms and is thought to reflect the refractoriness of axons of cortical interneurons subliminally activated by a subthreshold CS [Fisher et al., 2002] and/or extrasynaptic GABA tone [Stagg et al., 2011]. The second one observed at ISI=2.5/3 ms accounted for synaptic inhibition mediated by the GABAA receptors [Fisher et al., 2002; Peurala et al., 2008; Roshan et al., 2003]. The ISI duration was fixed at 3 ms in the current study as this ISI consistently demonstrates a clear peaks of inhibition [Cirillo and Byblow, 2016; Vucic et al., 2009]. However, future studies should address whether CS Intensity and ISI duration can affect SICIMmax modulation.”

The introduction will benefit with the logic as to why Opie and Semmler 2014 tested SICI in an active condition. In other words, measuring SICI-Meptest during activation is well-known to reduce the inhibitory effect. Should the prediction be the same for SICI-Mmax?
RESPONSE:  SICIMEPtest is well-known to reduce the inhibitory effect when assessed during muscle activation but importantly, this decrease has been shown to be specific according to the level of contraction tested [Hendy et al., 2019b; Rantalainen et al., 2013].
In their study, Opie & Semmler (2014) reported an increase in inhibition in response to larger MEPtest/Mmax ratios. Importantly, larger MEPtest/Mmax ratios were obtained by increasing the TS Intensity (110%, 120%, 130%, 140% and 150% of MT) during one active muscle state (5% of MVC).
Based on these findings, we expected that the SICIMmax modulation would be modulated likewise (i.e. increase in inhibition in response to larger MEPtest/Mmax ratios) but, at this time, with larger MEPtest/Mmax ratios obtained by increasing the level of contractions. By considering this prediction as well as Opie & Semmler (2014) findings, we also hypothesized that SICIMmax would be highly dependent upon the proportion of motoneurones activated by single-pulse TMS, regardless of the conditions leading to MEPtest increase.

The paragraph concerned in the introduction is now:

Line 64-89: “Taking into account this important point, Lackmy & Marchand-Pauvert (2010) assessed SICIMmax by calculating the difference between the mean MEPcond and the mean MEPtest then expressed as a percentage of Mmax (maximal compound muscle action potentials [M-wave]). Mmax is evoked when all the motor axons to the target muscle are activated simultaneously by a supramaximal peripheral nerve stimulation. SICIMmax normalisation has the methodological benefit to consider the proportion of spinal motoneurones activated by the TS. Interestingly, Lackmy & Marchand-Pauvert (2010) found increased SICIMmax with increasing TS intensities (ie increased MEPtest/Mmax ratio) [Lackmy and Marchand-Pauvert, 2010]. However, this work has only focused on resting hand muscle whereas muscle activation is known to decrease the amount of SICIMEPtest [Ortu et al., 2008; Ridding et al., 1995]. To fill this gap, Opie & Semmler (2014) tested subsequently whether SICIMmax (calculated with the MEPtest expressed relative to Mmax) was influenced when the muscle was voluntary activated. Contrary to SICIMEPtest, they found that the amount of SICIMmax during active state increased concomitantly with an increase in MEPtest/Mmax ratio [Opie and Semmler, 2014]. Importantly, larger MEPtest/Mmax ratios were obtained by increasing the TS Intensity (from 110% until 150% of the motor threshold) during the active state. However, in this study, active muscle condition was assessed with only one low level of contraction (5% of maximal voluntary contraction (MVIC)), whereas SICIMEPtest has been shown to vary with increasing levels of muscle contraction and MEPtest amplitude [Garry and Thomson, 2009; Hendy et al., 2019a; Ortu et al., 2008; Rantalainen et al., 2013; Zoghi and Nordstrom, 2007]. It is not yet known whether SICIMmax is modulated when assessed at higher levels of muscle contraction.

The aim of the present study was to broaden current knowledge of the necessity to consider the proportion of spinal motoneurones activated by single- and paired-pulse TMS in different muscle states. We quantified SICI with varying MEPtest amplitudes modulated actively (with four levels of muscle contractions) and passively (with two TS intensities of stimulation). The hypothesis was that SICIMmax would increase in response to larger MEPtest/Mmax ratios obtained by increasing the level of contractions. By considering this prediction as well as Opie & Semmler (2014) findings, we also hypothesized that the amount of SICIMmax would be highly dependent upon the proportion of motoneurones activated by single-pulse TMS, regardless of the conditions leading to MEPtest increase.”

Ortu et al;https://doi.org/10.1113/jphysiol.2008.158956 found that muscle activity effects on SICI were stronger when the CS intensity was larger (80 and 90% rMT). How do the results seen here take into account the differences found across CS intensity found in Ortu et al? Alternatively, could it be that 70% CS is only marginally effective at suppressing I-waves regardless of the condition in which it is probed? It would be important test whether a more effective CS intensity (either 80 or 90%) on the MEP would also show the same/different response seen in the SICI-Mmax at 90% CS.
RESPONSE: Actually, Ortu et al. (2008) found that “at rest the best inhibition was obtained delivering a S1 of 90% AMT while during contraction the most efficacious S1 was 70% AMT.” Similar results have been found in other studies. For example, Zoghi et al. (2003) compared SICI at ISI = 3 ms at rest and during muscular activity. They found that when the intensity of the TS was adapted to match the size of the MEPtest in the resting and active states, a CS of 70% aMT induced the same amount of inhibition in the two different conditions.
Moreover, in the study of Ortu et al., the aMT was calculated for each level of muscle contractions “for 10% MVC, the active motor threshold (AMT10%) was defined, as noted above, as the lowest stimulus intensity (% MSO) required to produce MEPs greater than 200 μV in at least 5 out of 10 consecutive stimuli. For 25% and 50% MVC, the AMT (AMT25% and AMT50%, respectively) was defined as the minimum stimulus intensity that produced at least five MEPs from 10 consecutive trials” and we followed exactly the same procedure in our study.

Therefore, the methodology used in the present study takes into account the findings of Ortu et al. (2008) and it appears that CS 70 %rMT (which approximately equates to 80/90 %aMT [Chen et al., 1998]) for resting condition or CS 70 %aMT for active conditions is effective at suppressing later I-waves.  Moreover, as mentioned in the previous Response, a recent study has demonstrated that the CS Intensity (60%, 70% or 80% aMT) did not alter the modulation of SICIMEPTest assessed at different muscle contraction levels (Hendy et al. 2019).
However, we completely agree with Reviewer #1 that it will be important to test how different CS Intensity (higher or lower) would affect the SICIMmax in active and resting states. This has been stated in the limitations section:

Line 450-451: “However, future studies should address whether CS Intensity and ISI duration can affect SICIMmax modulation”

Reviewer 3 Report

TITLE: Initially, upon reading the title, I thought this would be a review paper. Perhaps a title that outlines the purpose of the study or a key finding of the study would gain more interest from targeted readers. Something like “Effect of voluntary contraction level, test stimulus intensity, and normalisation procedures on short-interval intracortical inhibition”.

GENERAL FORMATTING:

Search for inconsistent spaces (e.g. line 145 there is a large gap between trials and in), and other spacing inconsistencies throughout.

Inconsistent use of capital letters – e.g. in the results section, sometimes TS ‘I’ntensity, sometimes TS ‘i’ntensity. Sometimes ‘L’evels of contraction, sometimes ‘l’evels of contraction. Resolve all inconsistencies.

Reference list has inconsistencies as well with capitalisation (e.g. ref #9) and it is not necessary to include the websites when all journal info is provided (e.g. #3, #8 etc).

ABSTRACT:

General comments: The abstract is missing some key methodological and statistical information. Who were the participants (n, age, gender)? What were the levels of muscle contractions? What intensities of test TMS stimulation? What muscle was tested, in which side of the body? Was handedness accounted for? Was only force controlled (as %MVIC) or was pre-stimulus sEMG also accounted for – we know the two are related, but recruitment strategies can change to still produce the same target force. Include statistical output to help interpretation of results.

Line 16: When introducing compound muscle action potential, it would be good to add M-wave in parentheses, and then in line 18 when Mmax is introduced, you can refer to it as Maximal M-wave (Mmax). Mmax was not previously abbreviated.

Line 24: This is a far stretch of a conclusion. Please be specific – in THIS population, in THIS muscle, under THESE conditions…. Your statement is too broad and makes the assumption that all muscles behave similarly, which is functionally and physiologically inaccurate.

INTRODUCTION:

Line 31: within *the* primary motor cortex.

Line 33-34: The first study to show that SICI was reduced following strength training was “Strength Training Reduces Intracortical Inhibition” by Weier, Pearce and Kidgell (2012). This would be an important citation here.

34-35: Contrasting results – this is important and needs to be acknowledged!!! WHY are they contrasting? There are MANY studies that have shown no change or even increases following strength training. The one consistency, though, is that if the strength training is sufficiently novel or neurologically challenging, the reduction in SICI is consistent and reproducible, and more closely aligns with that of skill acquisition, rather than strength increases per se. See the work of Michael Leung and colleagues.

Line 38: evaluate inhibitory circuits *within the M1*.

Line 49: previous studies have shown that in [the-delete] resting hand muscles

Line 56: as evidence*d* by epidural recordings

Line 69-70: Reword – this is an awkward sentence. Also should be ‘resting hand muscle’

Line 83-85: Write hypothesis in past tense.

MATERIALS AND METHODS:

Line 97: Custom-built*

Line 108: Awkward wording, revise.

Line 109: Each contraction levels?

Line 112: for *a* few seconds

Line 114-115: Please further explain the randomisation as this isn’t clear. I.e. did all stimuli at rest happen at once, then all at 10%MVIC happen at once, all at 20% happen at once, all at 30% happen at once, but these were counterbalanced whereby one participant might have started with 30%, then 10%, then 20%, then rest? Much more detail and clarity required around what exactly was randomised and how. Currently this cannot be replicated.

General experimental procedure – Was rmsEMG recorded during MVIC? This is an important cross-reference because whilst your force level may be controlled as a % of MVIC, the contractile behaviour of the muscle/s (including synergistic activity of other muscles) that contribute to the force may differ, especially during a long and repetitive protocol such as this. It is important to show that not only was force controlled, but the rmsEMG as a % of rmsEMG during MVIC also needs to be consistent throughout the protocol. There are also issues methodologically with the order of testing. Given that there is testing under resting conditions AND active conditions (x3), there is likely to be influence of the active conditions on the output from resting conditions via post-activation potentiation mechanisms. All resting conditions, including resting Mmax should have been done first, followed by MVIC, then at each contraction level the Mmax should have been re-recorded, and the TS and CS stimuli randomised.

Electrical stimulation protocol – It is not clear what condition the muscle was in during Mmax testing. It appears as though it was resting, but it is not mentioned. This is perhaps the most crucial aspect of the entire study because the fundamental argument here is that you are trying to control for background CMAP (influenced by different voluntary contraction levels). Therefore, Mmax should have been measured under all conditions (rest, 10% MVIC, 20% MVIC, 30% MVIC) and resulting MEPs normalised accordingly. Was this done? It is not explained here in the methods! à  I can see in figure 1, M-waves are displayed for all conditions. PLEASE clarify this in the methodology!!!

TMS protocol –

Line 139 – given intensity? What intensity was it? How was it determined?

Line 140 – wore / *worn*

Figure 1 – The y-axis scale in 1B and 1C in resting conditions is 0.2mV compared to 1mV for all others. Either make the y-axis scale consistent or make a note in the figure caption so that it is absolutely clear to the reader.

Statistical analyses – Normality and homogeneity of variance testing is mentioned, but it is not noted anywhere in the results whether all assumptions were met for all variables.

Line 196: the values are just written as mean +/- SD, not in parentheses? Just say that the values are reported as means +/- SD.

Line 198: You have accounted for rmsEMG/Mmax ratios but not respective pre-stim rmsEMG with the rmsEMG of the MVIC. This is more important than Mmax here.

Line 201: Were separate analyses conducted to separate active conditions from resting? Surely these are different questions?

RESULTS:

Table 1: In this table, include the mean rmsEMG for rest, 10%, 20%, 30% as a separate row under MT (%MSO) and as well under Mmax. rmsEMG acquired during peak MVIC would be ideal so you can demonstrate a % of max rmsEMG that hopefully corresponds to a similar % force (e.g. at 10% MVIC, you would hope this also represents close to 10% rmsEMG during MVIC). If rmsEMG doesn’t increase linearly from 0-10-20-30, you can postulate that force producing strategy differed, perhaps due to fatigue from repeated and sustained contractions. I don’t think this will be an issue though, given that there were no significant differences in Mmax, but there was a main effect for contraction level as shown in Fig 2A. It would just be helpful to show the raw rmsEMG data.

Line 238, 256 and others: Should be post hoc analyses*

Figure 2: Firstly, I would separate A out from the other 3. It tells a totally different story.

For B, C and D, it is unnecessary to have all different coloured bars when you have clearly labelled x-axes. Leave it as white = 120% MT, and a secondary option (plain black, plain grey, the diagonal lines – pick one) = 130% MT and keep all bars consistent. These graphs are too busy.

Line 269-270: This doesn’t make sense. Please revise the sentence.

Line 271: “for to predict”

DISCUSSION

Line 296-297: This wasn’t actually assessed here, though, was it?

Line 307: “fixed at 1mV amplitude in the majority of these studies” this does not fit here.

Line 310: “are the reflect”

Line 336: “makes *it* possible to”

Methodological recommendations: This is a stretch. The current study was completed acutely, and attempting to apply the same methodological approach following an intervention. Given the inconsistencies reported in all TMS assessments, including MEP amplitude, Mmax amplitude, MT, SICI, SP etc, it is important that these assessments are completed at the end of any intervention and not assumed to be predictable based on the present findings. Whilst Mmax was not different in THIS muscle, in THESE people between conditions, Mmax absolutely has been shown to be altered in response to various acute and chronic interventions, thus repeating testing is vital! Reconsider your methodological recommendations accordingly and perhaps tone this down a little.

Author Response

We would like to thank the Reviewers for taking the time to review our manuscript. We believe the comments and suggestions received will significantly strengthen the article. Hence, we have endeavored to address each comment as both a point-by-point response below, and within the updated version of the manuscript. In addition, a native English speaker edited our revised manuscript. 

TITLE: Initially, upon reading the title, I thought this would be a review paper. Perhaps a title that outlines the purpose of the study or a key finding of the study would gain more interest from targeted readers. Something like “Effect of voluntary contraction level, test stimulus intensity, and normalisation procedures on short-interval intracortical inhibition”.

RESPONSE: Following the Reviewer #2 concern, the title has been modified in order to give a better account of the key findings of the study:

Line 2-4: « Influence of voluntary contraction level, test stimulus intensity, and normalization procedures on the evaluation of short-interval intracortical inhibition »

GENERAL FORMATTING:

Search for inconsistent spaces (e.g. line 145 there is a large gap between trials and in), and other spacing inconsistencies throughout.
RESPONSE: We thank Reviewer #2 for pointing out these inconsistent spaces that has been corrected.

Inconsistent use of capital letters – e.g. in the results section, sometimes TS ‘I’ntensity, sometimes TS ‘i’ntensity. Sometimes ‘L’evels of contraction, sometimes ‘l’evels of contraction. Resolve all inconsistencies.
RESPONSE: The manuscript has been thoroughly revised accordingly.

Reference list has inconsistencies as well with capitalisation (e.g. ref #9) and it is not necessary to include the websites when all journal info is provided (e.g. #3, #8 etc).
RESPONSE: The reference list has been checked for inconsistencies with capitalization and websites references have been removed.

ABSTRACT:

General comments: The abstract is missing some key methodological and statistical information. Who were the participants (n, age, gender)? What were the levels of muscle contractions? What intensities of test TMS stimulation? What muscle was tested, in which side of the body? Was handedness accounted for? Was only force controlled (as %MVIC) or was pre-stimulus sEMG also accounted for – we know the two are related, but recruitment strategies can change to still produce the same target force. Include statistical output to help interpretation of results.
RESPONSE: We agree with Reviewer#2 that the abstract is missing some key information. Therefore, an effort has been made to provide this information to the reader (see below).
However, please note that some of the requested information (otherwise detailed in the text) has not been added to the abstract because of the constraint on the number of words (i.e. 200 words, which is too short for such details).

Line 16: When introducing compound muscle action potential, it would be good to add M-wave in parentheses, and then in line 18 when Mmax is introduced, you can refer to it as Maximal M-wave (Mmax). Mmax was not previously abbreviated.
RESPONSE: Following Reviewer #2 suggestions, these elements have been added (see following comment below).

Line 24: This is a far stretch of a conclusion. Please be specific – in THIS population, in THIS muscle, under THESE conditions…. Your statement is too broad and makes the assumption that all muscles behave similarly, which is functionally and physiologically inaccurate.
RESPONSE: We have considered the Reviewer’s concerns about the broad statement used in the conclusion. The last sentence has been removed and the other sentence has been changed.

Line 14-31: “Short-interval intracortical inhibition (SICI) represents an inhibitory phenomenon acting at the cortical level. However, SICI estimation is based on the amplitude of a motor-evoked potential (MEP), which depends on the discharge of spinal motoneurones and the generation of a compound muscle action potential (M-wave). In this study, we underpinned the importance of taking into account the proportion of spinal motoneurones activated or not when investigating SICI of the right flexor carpi radialis (normalization with maximal M-wave (Mmax) and MEPtest, respectively) in 15 healthy subjects. We probed SICI changes according to various MEPtest amplitude that was modulated actively (four levels of muscle contractions: rest, 10%, 20% and 30% of maximal voluntary contraction (MVC)) and passively (two intensities of Test TMS stimulation: 120 and 130% of motor thresholds). When normalized to MEPtest, SICI remained unchanged according to the stimulation intensity and only decreased at 30% of MVC when compared to rest. However, when normalized to Mmax, we provided first evidence for a strong individual relationship between SICI and MEPtest, which was ultimately independent from experimental conditions (muscle states and TMS intensities). Under similar experimental conditions, it is thus possible to individually predict SICI from a specific level of corticospinal excitability in healthy subjects.”

INTRODUCTION:

Line 31: within *the* primary motor cortex.
RESPONSE: We thank Reviewer #2 for pointing out this mistake that has been corrected:

Line 33-34: The first study to show that SICI was reduced following strength training was “Strength Training Reduces Intracortical Inhibition” by Weier, Pearce and Kidgell (2012). This would be an important citation here.
RESPONSE: We agree with Reviewer #2 that this important study was missing in the previous version of the manuscript. The reference has been added.

34-35: Contrasting results – this is important and needs to be acknowledged!!! WHY are they contrasting? There are MANY studies that have shown no change or even increases following strength training. The one consistency, though, is that if the strength training is sufficiently novel or neurologically challenging, the reduction in SICI is consistent and reproducible, and more closely aligns with that of skill acquisition, rather than strength increases per se. See the work of Michael Leung and colleagues.
RESPONSE: We agree with Reviewer #2 that SICI the reduction of SICI following strength training is task-dependent (Siddique et al. 2020). However, the aim of this methodological paper was not to discuss about specific SICI modulation following strength-training program. In order to clarify the example of SICI modulation following intervention, the sentence has been modified:

Line 37-39: “For example, the reduction of intracortical inhibition within M1 is a crucial part of neural adaptation following acute and multi-session challenging strength-training programs [2–4].with, however, some contrasting results [5,6]

Line 38: evaluate inhibitory circuits *within the M1*.
RESPONSE: This has been added.

Line 41-42: “Short-interval intracortical inhibition (SICI) is a well-established paired-pulse measure to evaluate inhibitory circuits within the M1 area.”

Line 49: previous studies have shown that in [the-delete] resting hand muscles
RESPONSE: This has been deleted.

Line 56: as evidence*d* by epidural recordings
REPONSE: This has been corrected.

Line 69-70: Reword – this is an awkward sentence. Also should be ‘resting hand muscle’
RESPONSE: This sentence has been reworded as follow:

Line 74-75: “However, this work has only focused on resting hand muscle whereas muscle activation is known to decrease the amount of SICIMEPtest

Line 83-85: Write hypothesis in past tense.
RESPONSE: Hypothesis has been re-written in past tense and changed according to Reviewer#1 concern.

Line 95-99: “The hypothesis was that SICIMmax would increase in response to larger MEPtest/Mmax ratios obtained by increasing the level of muscle contraction. By considering this prediction as well as Opie & Semmler (2014) findings, we also hypothesized that the amount of SICIMmax would be highly dependent upon the proportion of motoneurones activated by single-pulse TMS, regardless of the conditions leading to MEPtest increase.”

MATERIALS AND METHODS:

Line 97: Custom-built*
RESPONSE: This has been corrected.

Line 108: Awkward wording, revise.
RESPONSE: This sentence has been reworded as follow:

Line 122-123: “This value was then used to calculate the muscle contraction levels (i.e., 10, 20 and 30 % MVIC) tested in the current study.”

Line 109: Each contraction levels?
RESPONSE: The whole paragraph about randomization (which includes this sentence) has been reworded (see below).

Line 112: for *a* few seconds
RESPONSE: We thank Reviewer #2 for pointing out this mistake that has been corrected.

Line 114-115: Please further explain the randomisation as this isn’t clear. I.e. did all stimuli at rest happen at once, then all at 10%MVIC happen at once, all at 20% happen at once, all at 30% happen at once, but these were counterbalanced whereby one participant might have started with 30%, then 10%, then 20%, then rest? Much more detail and clarity required around what exactly was randomised and how. Currently this cannot be replicated.
RESPONSE: To address this concern, the randomization used in the current study has been clarified:

Line 125-137: “For each levels of contraction (rest, 10%, 20% and 30 %MVIC), three Mmax traces, twelve single-pulse MEPs (MEPtest) and twelve paired-pulse MEPs (MEPcond) delivered with each of the TS intensities (120% and 130% of MT) were recorded per subject. The order of the different levels of contraction was randomized across participants. Single-pulse and paired-pulse TMS measures obtained with the two TS intensities were performed in different recording blocks and the order of the blocks was randomized order across participants. The inter-trial intervals between single-pulses was at least 3 s [22]. During active conditions, the mean duration of the recording was set at 500 ms and participants were asked to relax after the stimulation for few seconds before the next trial and contracting again. This was done to avoid fatigue generated by holding the level of contraction across all consecutive trials.”

General experimental procedure – Was rmsEMG recorded during MVIC? This is an important cross-reference because whilst your force level may be controlled as a % of MVIC, the contractile behaviour of the muscle/s (including synergistic activity of other muscles) that contribute to the force may differ, especially during a long and repetitive protocol such as this. It is important to show that not only was force controlled, but the rmsEMG as a % of rmsEMG during MVIC also needs to be consistent throughout the protocol. There are also issues methodologically with the order of testing. Given that there is testing under resting conditions AND active conditions (x3), there is likely to be influence of the active conditions on the output from resting conditions via post-activation potentiation mechanisms. All resting conditions, including resting Mmax should have been done first, followed by MVIC, then at each contraction level the Mmax should have been re-recorded, and the TS and CS stimuli randomised.
REPONSE: The rmsEMG during MVIC was recorded but not analyzed. Following Reviewer’s suggestion, we have extracted the rmsEMG obtained during the 3-s hold of the participant’s MVIC. Overall, it appears that when normalized, the rmsEMG of the MEP data (see Table below) are quite smaller than the ones obtained with the different force levels. However, it is known that the isometric surface EMG-force relationship is nonlinear. For example, Woods & Bigland-Ritchie (1983) reported a nonlinear EMG-force relationship for upper limb muscles. Particularly, nonlinearities occurred when forces were below 40% of MVIC, then changing to a linear relationship at levels of force above 40% of MVIC [Woods and Bigland-Ritchie, 1983].
Importantly, like observed for the rmsEMG/Mmax, the %rmsEMG-MVIC increased when increasing the level of contraction.

Subject

rmsEMG-MVIC
(mV)

Rest
(%rmsEMG-MVIC)

10% MVIC
(%rmsEMG-MVIC)

20% MVIC
(%rmsEMG-MVIC)

30% MVIC
(%rmsEMG-MVIC)

S01

2.652

0.047

2.974

4.991

8.350

S02

0.711

0.176

4.624

9.254

14.495

S03

0.383

0.398

8.008

15.557

21.343

S04

0.975

0.439

4.605

10.430

14.729

S05

2.147

0.108

6.253

9.671

13.108

S06

0.598

0.300

10.367

15.782

23.925

S07

1.622

0.077

5.484

7.948

16.162

S08

0.372

0.445

13.926

19.251

26.926

S09

0.382

0.435

12.949

18.614

29.793

S10

0.402

0.422

10.226

18.960

21.297

S11

1.308

0.104

6.414

14.285

15.525

S12

0.754

0.286

7.980

11.824

17.159

S13

0.421

0.563

23.343

30.900

35.328

S14

0.460

0.393

19.198

38.321

50.353

S15

0.744

0.431

8.353

15.059

20.964

Mean

0.929

0.308

9.647

16.056

21.964

SD

0.705

0.165

5.672

8.744

10.509

Since we cannot exclude that a synergist muscle also contributed to the force production, it has been acknowledged in the Limitation section:

Line 440-444: “Finally, only one forearm muscle was tested, and we did not consider 1) antagonist muscle and 2) synergistic muscles that are also relevant for force production without a concomitant increase in the EMG recorded in the FCR agonist muscle. At this stage, the results cannot be generalized to all muscle groups and deserve further investigations.”

Moreover, concerning the methodological issues with the order of testing, we agree that the post-activation potentiation phenomenon can induce a modulation of the Mmax amplitude and therefore a change in the EMG signals. However, all the measures were performed after a warm-up and MVIC (with a long period of break after the MVIC to minimize the potentiation effect that is known to occur for a »30 s. period). Importantly, three Mmax were recording and used for the MEPs and rmsEMG normalization for each level of contraction. Moreover, during active conditions, participants were asked to relax after the stimulation for few seconds before the next trial and contracting again. This was done to avoid fatigue generated by holding the level of contraction across all consecutive trials.  Finally, we believe that starting with all resting conditions, followed by MVIC, then at each contraction levels would not solve the potentiation issue since we would have induced non-potentiated responses at rest but potentiated responses for submaximal contraction conditions.  Therefore, a complete randomization of all conditions for a given level of contraction, with Mmax normalization obtained for each level of contraction appeared to us the best methodological approach. Thus, the latest measurements carried out for a given level of contraction vary amongst participants, which limits measurement biais.

Electrical stimulation protocol – It is not clear what condition the muscle was in during Mmax testing. It appears as though it was resting, but it is not mentioned. This is perhaps the most crucial aspect of the entire study because the fundamental argument here is that you are trying to control for background CMAP (influenced by different voluntary contraction levels). Therefore, Mmax should have been measured under all conditions (rest, 10% MVIC, 20% MVIC, 30% MVIC) and resulting MEPs normalised accordingly. Was this done? It is not explained here in the methods! I can see in figure 1, M-waves are displayed for all conditions. PLEASE clarify this in the methodology!!!
RESPONSE: In the method section, we first specified that Mmax has been tested for all contraction levels: “Each contraction levels were considered as one block of trials (4 blocks in total: rest, 10, 20 and 30 % MVIC). One block comprised three Mmax traces per subject.”  
However, this may have been unclear and following Reviewer’s concern we clarified in the Methods section:

Line 125-128: “For each levels of contraction (rest, 10%, 20% and 30 %MVIC), three Mmax traces, twelve single-pulse MEPs (MEPtest) and twelve paired-pulse MEPs (MEPcond) delivered with each of the TS intensities (120% and 130% of MT) were recorded per subject.”

TMS protocol –

Line 139 – given intensity? What intensity was it? How was it determined?
RESPONSE: This given intensity was individualized for each participant and was defined as the intensity able to induce clear MEPs on EMG recordings. In practice, the intensity was set at 40 %MSO and was adjusted for each participant.

Line 140 – wore / *worn*
RESPONSE: This has been corrected.

Figure 1 – The y-axis scale in 1B and 1C in resting conditions is 0.2mV compared to 1mV for all others. Either make the y-axis scale consistent or make a note in the figure caption so that it is absolutely clear to the reader.
REPONSE: We agree with Reviewer#2 that changing the y-axis scale can be disturbing for the reader. However, the MEPs amplitude were highly different between resting and active conditions making the traces unreadable with a consistent scale. Therefore, we have followed Reviewer’s second recommendation and the following note has been added in the figure caption:

Line 213-214: “Note that the y-axis scale for MEPs amplitude differs between resting and active conditions (i.e. 0.2 mV vs. 1 mV) for a better graphical visualization”

Statistical analyses – Normality and homogeneity of variance testing is mentioned, but it is not noted anywhere in the results whether all assumptions were met for all variables.
RESPONSE: This has been clarified.

Line 217-218: “Normality of the data distributions was verified using the Shapiro-Wilk test and the assumption of normality was not violated for any of the data.”

Line 221: “Uncorrected degrees of freedom and corrected p values for multiple comparisons are reported in the results section.”

Line 196: the values are just written as mean +/- SD, not in parentheses? Just say that the values are reported as means +/- SD.
REPONSE: This has been changed as follow:

Line 225: “Values are reported as mean ± SD.”

Line 198: You have accounted for rmsEMG/Mmax ratios but not respective pre-stim rmsEMG with the rmsEMG of the MVIC. This is more important than Mmax here.
RESPONSE: Since the Mmax have been recorded in similar conditions than rmsEMG (timing and contraction levels), we believed that this normalization is correct for the purpose of the current study. Normalizing EMG activity by Mmax would allow us to account for various potential changes such as fatigue, but also to focus on central component of such evoked potentials. Indeed, motor potential are recorded through surface EMG at peripheral level and are therefore dependent upon the neuromuscular junction. This latter could be influenced by the level of voluntary contraction mainly because of intramuscular mechanisms. Therefore normalizing by Mmax is the best way to avoid any muscular mechanisms interfering within the EMG signal. Additionally, the maximal Mwave, by being evoked supramaximally and triggered externally, represent the full potential of the available motor units independently of the participant’s voluntary activation level.  The same rmsEMG/Mmax normalization and has been utilized in a previous similar study (Hendy et al. 2019).
A Table with the rmsEMG expressed in percentage of rmsEMG-MVIC has also been added, showing that it increased according to the level of muscle contraction.

Line 251-255: “Comparison of MT (%MSO), raw EMGrms for MEP data (mV), EMGrms expressed as a % of EMGrms obtained during MVIC; Mmax amplitude (mV) and raw EMGrms for Mmax data (mV) across Levels of muscle contraction (%MVIC). Data represent mean ± SD. The MT for the Rest condition was significantly higher than all the MT obtained for active conditions (10, 20 and 30% MVIC).”

Levels of muscle contraction (%MVIC)

Rest

10%

20%

30%

MT (%MSO)

40.87 ± 7.3 **

31.73 ± 8.1

31.53 ± 7.2

31.27 ± 7

rmsEMG MEP (mV)

1.97 ± 0.8

67.18 ± 28.2

113.58 ± 46.5

160.32 ± 64.5

rmsEMG (% rmsEMG-MVIC)

0.31 ± 0.2

9.65 ± 5.7

16.06 ± 8.7

21.96 ± 10.5

Mmax (mV)

6.97 ± 3.7

6.82 ± 3.8

6.60 ± 3.7

6.70 ± 3.9

rmsEMG Mmax (mV)

1.75 ± 0.6

65.04 ± 27.7

114.83 ± 61

160.73 ± 78.8

Line 201: Were separate analyses conducted to separate active conditions from resting? Surely these are different questions?
RESPONSE: The statistical analyses for TMS data (MEPtest/Mmax; SICIMEPtest and SICIMmax) were conducted with the main effect “Levels of contraction” that includes both the resting and active conditions. The objective of these analysis was not to separate the resting and active conditions but rather to modulate the MEPtest amplitude actively with different muscle states. However, with these analyses, if a main effect of “levels of contraction” was found, the post-hoc analyses carried out make it possible to investigate whether the resting and active conditions were significantly different.

RESULTS:

Table 1: In this table, include the mean rmsEMG for rest, 10%, 20%, 30% as a separate row under MT (%MSO) and as well under Mmax. rmsEMG acquired during peak MVIC would be ideal so you can demonstrate a % of max rmsEMG that hopefully corresponds to a similar % force (e.g. at 10% MVIC, you would hope this also represents close to 10% rmsEMG during MVIC). If rmsEMG doesn’t increase linearly from 0-10-20-30, you can postulate that force producing strategy differed, perhaps due to fatigue from repeated and sustained contractions. I don’t think this will be an issue though, given that there were no significant differences in Mmax, but there was a main effect for contraction level as shown in Fig 2A. It would just be helpful to show the raw rmsEMG data.
RESPONSE: We have reported in the Table 1 the mean rmsEMG (mV) for rest, 10%, 20%, 30% as a separate row under MT (%MSO) and the mean rmsEMG (mV) for rest, 10%, 20%, 30% as a separate row as well under Mmax. We have also reported in this Table the rmsEMG expressed in percentage of rmsEMG-MVIC.

Levels of muscle contraction (%MVIC)

Rest

10%

20%

30%

MT (%MSO)

40.87 ± 7.3 **

31.73 ± 8.1

31.53 ± 7.2

31.27 ± 7

rmsEMG MEP (mV)

1.97 ± 0.8

67.18 ± 28.2

113.58 ± 46.5

160.32 ± 64.5

rmsEMG (% rmsEMG-MVIC)

0.31 ± 0.2

9.65 ± 5.7

16.06 ± 8.7

21.96 ± 10.5

Mmax (mV)

6.97 ± 3.7

6.82 ± 3.8

6.60 ± 3.7

6.70 ± 3.9

rmsEMG Mmax (mV)

1.75 ± 0.6

65.04 ± 27.7

114.83 ± 61

160.73 ± 78.8

We conducted a repeated-measure ANOVA with the Levels of contraction4 as a within-subject factor on raw mean rmsEMG obtained for rest, 10%, 20% and 30% of MVIC. The main effect of Level of contraction was significant (F(3,42) = 78.04; p < .001) and we found the same result than the rmsEMG/Mmax normalization with a lower EMGrms/Mmax ratio for the rest condition compared to the 10% MVIC (p < .001), 10% MVIC compared to 20% MVIC (p =.001) and 20% MVIC compared to the 30% MVIC (p = .001).
However, as said in a previous comment, the isometric surface EMG-force relationship is thought to be nonlinear (but rather be curvilinear) for upper limb muscle when tested forces as below 40% of MVIC [Woods and Bigland-Ritchie, 1983].
We believe that the rmsEMG normalization chosen has no consequence on the MEP and SICI results and their interpretation, because 1) Mmax was recorded for each levels of contraction, 2) there were no significant differences in Mmax, but there was a main effect for contraction level across the participants as highlighted by the Reviewer.

Line 238, 256 and others: Should be post hoc analyses*
RESPONSE: These mistakes have been corrected.

Figure 2: Firstly, I would separate A out from the other 3. It tells a totally different story.
RESPONSE: This has been done. The RMSEMG/Mmax ratios plotted according to the Levels of contraction is now a separate Figure (Figure 2) from the TMS results.

Figure 3: RMSEMG/Mmax ratios pooled for MEPtest (unconditioned) and MEPcond (conditioned) and the two Test Stimulus (TS) Intensities (120% and 130% of motor threshold (MT)).

For B, C and D, it is unnecessary to have all different coloured bars when you have clearly labelled x-axes. Leave it as white = 120% MT, and a secondary option (plain black, plain grey, the diagonal lines – pick one) = 130% MT and keep all bars consistent. These graphs are too busy.
RESPONSE: Following this concern, the graphs are now more consistent with only white (120 %MT) and black (130% MT) colors.

Figure 4: (A) MEPtest/Mmax ratios according to the TS Intensity and the Levels of contraction. (B) SICI expressed in percentage of MEPtest (SICIMEPtest) according to the TS Intensity and the Levels of contraction. (C) SICI expressed in percentage of Mmax (SICIMmax) according to the TS Intensity and the Levels of contraction. * p < .05; ** p < .001

Line 269-270: This doesn’t make sense. Please revise the sentence.
RESPONSE: The sentence has been corrected.

Line 307-308: “SICIMEPtest (Figure 5A) and SICIMmax (Figure 5B) plotted according to the MEPtest/Mmax values pooled into different bins.”

Line 271: “for to predict”
RESPONSE: This has been corrected:

Line 309-310: “First, a simple linear regression was calculated on mean bins values in order to predict the SICIMEPtest based on the MEPtest/Mmax ratio (Figure 5C).”

DISCUSSION

Line 296-297: This wasn’t actually assessed here, though, was it?
RESPONSE: We agree with Reviewer#2 that the linearity of the corticospinal excitability increase has not been actually assessed. The sentence has been clarified by removing the word “linearly”:

Line 335-336: “As expected, corticospinal excitability increased with increasing muscle contraction and TS intensity [26,27].”

Line 307: “fixed at 1mV amplitude in the majority of these studies” this does not fit here.
RESPONSE: This has been moved:

Line 345-346: “One explanation is that during high levels of voluntary muscle contraction, the excitability of the spinal motoneurones pool is increased and the later indirect-waves (called I3) generated by the TS (fixed to induce 1mV MEP amplitude in the majority of previous studies) no longer contribute to the production of the MEPtest [28]”

Line 310: “are the reflect”
RESPONSE: This has been corrected:

Line 341-344: “While direct D-wave is the first latency response originating from the activation of the pyramidal tract axons, the later indirect I-waves, numbering 3 waves (I1, I2, I3), which are thought to result from the pyramidal neurons activation [29]”

Line 336: “makes *it* possible to”
RESPONSE: This has been added.

Methodological recommendations: This is a stretch. The current study was completed acutely, and attempting to apply the same methodological approach following an intervention. Given the inconsistencies reported in all TMS assessments, including MEP amplitude, Mmax amplitude, MT, SICI, SP etc, it is important that these assessments are completed at the end of any intervention and not assumed to be predictable based on the present findings. Whilst Mmax was not different in THIS muscle, in THESE people between conditions, Mmax absolutely has been shown to be altered in response to various acute and chronic interventions, thus repeating testing is vital! Reconsider your methodological recommendations accordingly and perhaps tone this down a little.
RESPONSE: The procedure we suggested in the “Methodological recommendations” was to avoid the time-consuming adjustment of TMS intensity during post-test measures. However, it did not exclude the rest of the classical measures and especially the Mmax recording in each condition and at each timing (pre- and post-tests). 

Following the Reviewer’s concern, we have also added:

Line 428-432: “It is important to keep in mind that these methodological recommendations are based on results obtained in a study completed acutely. Since Mmax does not always remain constant during the course of an experiment [Crone et al., 1999] and following strength-training [Nuzzo et al., 2016], it is crucial to repeat the testing for each experimental condition, including during post-test measurements.”

Round 2

Reviewer 2 Report

The resubmitted article has greatly improved from the previous version and the authors have satisfied my original concerns. Great improvements are noted especially within the introduction and data analysis sections. There are still a few minor grammatical errors to address, other than this, the article is in good shape.

Author Response

Comments and Suggestions for Authors

The resubmitted article has greatly improved from the previous version and the authors have satisfied my original concerns. Great improvements are noted especially within the introduction and data analysis sections. There are still a few minor grammatical errors to address, other than this, the article is in good shape.

RESPONSE: We would like to thank the Reviewer 2 for the positive comments and valuable previous comments and suggestions, which we believe have improved the quality of the paper.
We have carefully corrected the grammatical error and expression issues through the manuscript.

Reviewer 3 Report

The authors should be congratulated on the significant amount of work they've put into improving this paper. It is infinitely better than the original submission. 

The introduction provides a much stronger rationale and more relevant background for the study. 

The methods - which were of greatest concern to me - have been improved with much greater clarity to the reader - particularly around the randomisation processes and inclusion of pre-stim rmsEMG, as well as clarifying outcomes for tests of parametric test assumptions.

The results section is also much more comprehensive with the inclusion of rmsEMG.

The figures are much better!

You also should be commended on the revisions to methodological recommendations and the limitations. 

Comments requiring attention:

It is important to ensure that you're clear on what exactly that rms is indicative of - often it is just mentioned as rmsEMG without clarification on what it represents. E.g. in Figure 3 - is this the pre-stimulus rmsEMG in that 100ms epoch or...? Try to be explicitly clear throughout.

There are some absolutely minor expression issues throughout, including in the abstract. A very careful proof read is required but otherwise it reads very well. 

Author Response

Comments and Suggestions for Authors

  1. The authors should be congratulated on the significant amount of work they've put into improving this paper. It is infinitely better than the original submission. 
  2. The introduction provides a much stronger rationale and more relevant background for the study.
  3. The methods - which were of greatest concern to me - have been improved with much greater clarity to the reader - particularly around the randomisation processes and inclusion of pre-stim rmsEMG, as well as clarifying outcomes for tests of parametric test assumptions.
  4. The results section is also much more comprehensive with the inclusion of rmsEMG.
  5. The figures are much better!
  6. You also should be commended on the revisions to methodological recommendations and the limitations. 

RESPONSE: Thank you for the positive comments, we have highly appreciated the useful previous suggestions that have significantly strengthen the article.

Comments requiring attention:

  1. It is important to ensure that you're clear on what exactly that rms is indicative of - often it is just mentioned as rmsEMG without clarification on what it represents. E.g. in Figure 3 - is this the pre-stimulus rmsEMG in that 100ms epoch or...? Try to be explicitly clear throughout.

RESPONSE: We have clarified what exactly the EMGrms is indicative of.

Line 260-261 (Figure 3 caption): “EMGrms were recorded 100ms before the delivery of each TMS pulse.”

Line 244 (Table 1): “single- and paired-pulse TMS raw EMGrms (mV)”

  1. There are some absolutely minor expression issues throughout, including in the abstract. A very careful proof read is required but otherwise it reads very well. 

REPONSE: We have carefully proofread the abstract and the manuscript.